# Gre factors help *Salmonella* adapt to oxidative stress by improving transcription elongation and fidelity of metabolic genes

Sashi Kant[1], James Karl A. Till[1,2], Lin Liu[1,3], Alyssa Margolis[1,2], Siva Uppalapati[1], Ju-Sim Kim[1,3], Andres Vazquez-Torres [1,2,3]*

1 University of Colorado School of Medicine, Department of Immunology & Microbiology, Aurora, Colorado, United States of America, 2 University of Colorado School of Medicine, Molecular Biology Program, Aurora, Colorado, United States of America, 3 Veterans Affairs Eastern Colorado Health Care System, Denver, Colorado, United States of America

* Andres.Vazquez-Torres@cuanschutz.edu

**Data Availability Statement:** All relevant data are within the paper and its Supporting Information files.

## Abstract

Detoxification, scavenging, and repair systems embody the archetypical antioxidant defenses of prokaryotic and eukaryotic cells. Metabolic rewiring also aids with the adaptation of bacteria to oxidative stress. Evolutionarily diverse bacteria combat the toxicity of reactive oxygen species (ROS) by actively engaging the stringent response, a stress program that controls many metabolic pathways at the level of transcription initiation via guanosine tetraphosphate and the α-helical DksA protein. Studies herein with *Salmonella* demonstrate that the interactions of structurally related, but functionally unique, α-helical Gre factors with the secondary channel of RNA polymerase elicit the expression of metabolic signatures that are associated with resistance to oxidative killing. Gre proteins both improve transcriptional fidelity of metabolic genes and resolve pauses in ternary elongation complexes of Embden–Meyerhof–Parnas (EMP) glycolysis and aerobic respiration genes. The Gre-directed utilization of glucose in overflow and aerobic metabolism satisfies the energetic and redox demands of *Salmonella*, while preventing the occurrence of amino acid bradytrophies. The resolution of transcriptional pauses in EMP glycolysis and aerobic respiration genes by Gre factors safeguards *Salmonella* from the cytotoxicity of phagocyte NADPH oxidase in the innate host response. In particular, the activation of cytochrome *bd* protects *Salmonella* from phagocyte NADPH oxidase-dependent killing by promoting glucose utilization, redox balancing, and energy production. Control of transcription fidelity and elongation by Gre factors represent important points in the regulation of metabolic programs supporting bacterial pathogenesis.

## Introduction

Oxidative stress is one of the most potent effectors of the innate immune system [1–3]. Reactive oxygen species (ROS) generated by NADPH oxidase (NOX2) in the respiratory burst of phagocytic cells exert potent anti-*Salmonella* activity [4], damaging nucleic acids, amino acid residues, and metal prosthetic groups [5]. Considerable attention has been paid to the

**Funding:** This work was supported by a VA (Merit Grant BX0002073 to AVT), and NIH grants (R01AI54959 and R01AI136520 to AVT, and T32AI052066 to JT). The funders had no role in study design, data collection and analysis, decision to publish, or preparation of the manuscript.

**Competing interests:** The authors have declared that no competing interests exist.

**Abbreviations:** EMP, Embden–Meyerhof–Parnas; ETC, electron transport chain; LB, Luria–Bertani; M-MLV, Moloney murine leukemia virus; PBS, phosphate-buffered saline; PCI, phenol/chloroform/isoamyl; qRT-PCR, quantitative real-time PCR; ROS, reactive oxygen species; SNS, single-nucleotide substitution; SPI-2, Salmonella pathogenicity island-2; UHPLC, ultra-high-performance liquid chromatography.

antioxidant defenses that counteract the tremendous selective pressures of respiratory burst products. Effectors of the *Salmonella* pathogenicity island-2 (SPI-2) type III secretion system divert NOX2-containing vesicles away from phagosomes [6]. Iron-sequestering proteins prevent formation of genotoxic ferryl and hydroxyl radicals in the $Fe^{2+}$-mediated reduction of hydrogen peroxide ($H_2O_2$) [7]. ROS that reach intracellular *Salmonella* are detoxified by periplasmic superoxide dismutases, catalases and hydroperoxide reductases [8–10], or are scavenged by the low-molecular weight thiol glutathione (GSH) [11]. Despite all of these protective mechanisms, ROS are still able to damage the genome and proteome, necessitating restoration by DNA repair systems and thioredoxin [12,13]. In addition to these antioxidant defenses, recent investigations have uncovered metabolic reprogramming as a potent, yet still little understood mechanism in the adaptation of bacteria to the vigorous antimicrobial activity of ROS engendered in the innate host response [14]. *Salmonella* sustaining oxidative stress favor glycolysis, fermentation, and the reductive tricarboxylic acid cycle, while slowing down redox balancing in the electron transport chain (ETC) [14]. By doing so, this facultative intracellular pathogen not only diminishes the adventitious generation of superoxide anion in the ETC, but also allows for the vectorial delivery of electrons by the Dsb thiol-disulfide exchange system into oxidized quinones [14]. Diminishing energetic production in oxidative phosphorylation facilitates repair of oxidized periplasmic proteins, while still meeting energetic and redox balancing needs [14].

By recruiting RNA polymerase to the promoters of target genes, transcription factors such as Fnr, PhoP, SsrB, or alternative $\sigma^S$ and $\sigma^E$ sigma factors activate expression of antioxidant defenses [15–19]. *Salmonella* have also harnessed the regulatory activity that the nucleotide alarmone guanosine tetraphosphate (ppGpp) and the small protein DksA exert on RNA polymerase to adapt to NOX2-dependent cytotoxicity [20–22]. Direct interactions of DksA with RNA polymerase boost *Salmonella*'s antioxidant defenses by both controlling redox balance and stimulating SPI-2 gene transcription [21,23]. The regulation of redox balance by the stringent response illustrates the essentiality of metabolism in the adaptation to oxidative stress.

In addition to accommodating DksA, the secondary channel of RNA polymerase houses Gre factors, which share similarities with DksA in overall α-helical folding and a pair of conserved acidic residues at the flexible loop in the C-terminal coiled-coil domain [24]. In contrast to DksA proteins, which mostly control initiation of transcription [25], Gre factors regulate the elongation and initiation steps by coordinating a $Mg^{2+}$ cation in the active site of RNA polymerase, which catalyzes water-mediated endonuclease cleavage of nascent transcripts backtracked into the secondary channel [26]. Incorporation of incorrect ribonucleotides during mRNA synthesis also arrests transcription elongation, activating the Gre-dependent endonuclease activity of RNA polymerase [27]. In this context, proofreading of nascent RNA molecules via Gre factors helps maintain transcriptional fidelity [28].

Herein, we have tested whether the proofreading and pause-relieving activities of Gre factors promote resistance of *Salmonella* to oxidative stress. Our investigations demonstrate that the control of transcriptional fidelity and transcription elongation of central metabolic genes by Gre factors results in biosynthetic, energetic, and redox outputs that promote *Salmonella* fitness during periods of oxidative stress.

## Results

### Gre factors defend *Salmonella* against the oxidative products of the phagocyte NADPH oxidase

In the following investigations, we tested whether the 2 homologous Gre proteins encoded in the *Salmonella* chromosome participate in bacterial pathogenesis. Compared to wild-type

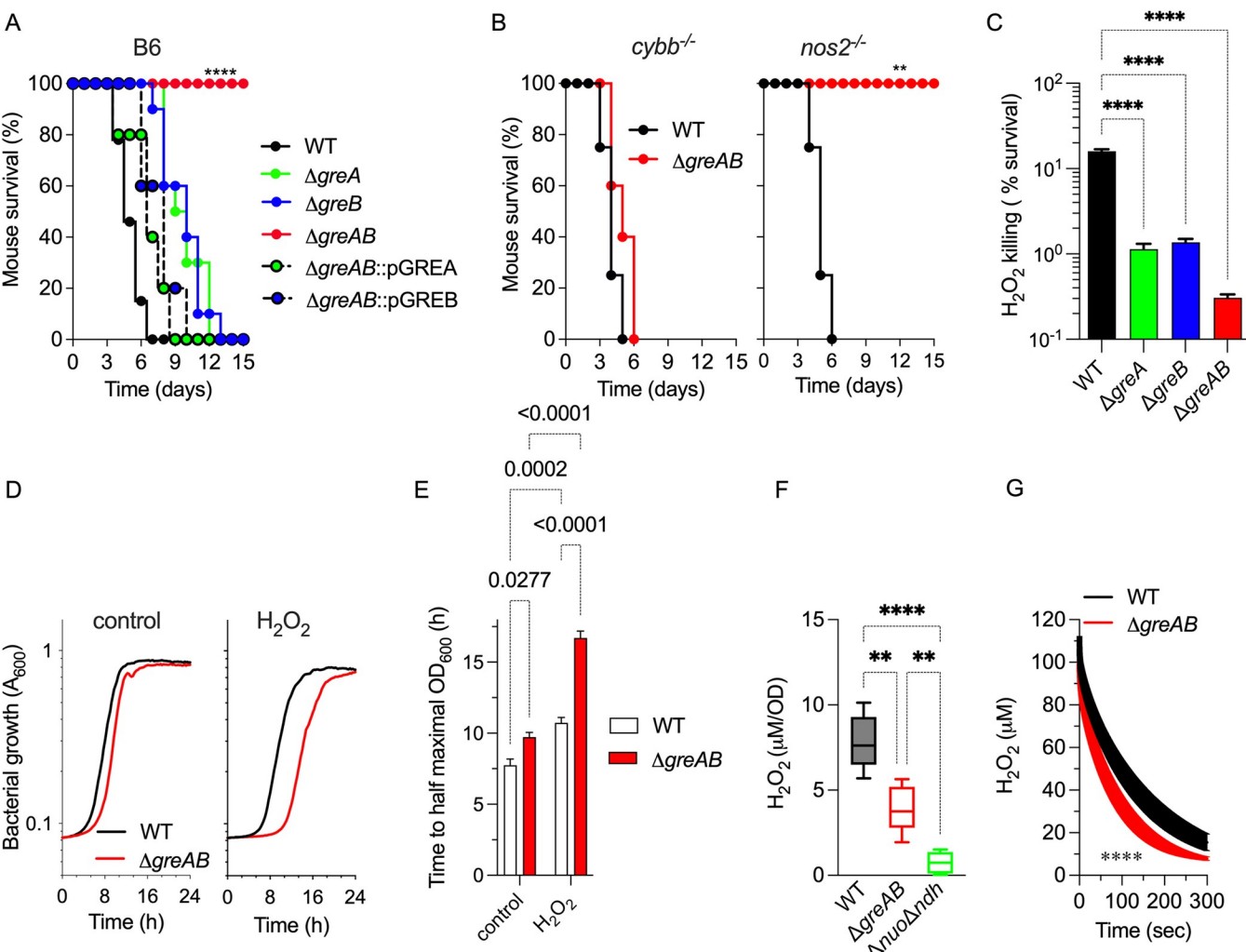

**Fig 1. Gre factors regulate resistance of *Salmonella* to the antimicrobial activity of NOX2.** Survival of C57BL/6 (A), *cybb$^{-/-}$* and *nos2$^{-/-}$* (B) mice after i.p. inoculation of 100 CFU of the indicated strains of *Salmonella* (*N* = 10). Statistical differences (*p* < 0.0001) were calculated by logrank analysis. The Δ*greAB* mutant was complemented with either *greA* or *greB* genes expressed from their native promoters in the low copy number pWSK29 plasmid (i.e., pGREA and pGREB, respectively). (C) Killing of *Salmonella* 2 h after treatment with 400 μM $H_2O_2$. Bacterial cultures were grown overnight in LB broth, diluted to $2 \times 10^5$ CFU/ml in PBS and treated for 2 h with 400 μM $H_2O_2$. Killing, expressed as percent survival compared to the bacterial burden at time zero, is the mean ± SD (*N* = 12–16); *p* < 0.0001 as determined by one-way ANOVA. (D) Growth of wild-type (WT) and Δ*greAB* mutant *Salmonella* in EG minimal medium (pH 7.0), at 37°C in the presence or absence of 200 μM $H_2O_2$ as measured by $OD_{600}$ in a Bioscreen plate reader. Data are shown as the mean (*N* = 11–12). (E) Time to reach half maximal $OD_{600}$ was calculated from curves in panel D. Data are the mean ± SD. *p* determined by two-way ANOVA. Endogenous $H_2O_2$ synthesis (F) and $H_2O_2$ consumption (G) by log phase *Salmonella* grown in MOPS-GLC medium (pH 7.2), at 37°C with shaking to an $OD_{600}$ of 0.25. $H_2O_2$ was measured polarographically in an APOLLO 4000 free radical analyzer. (G) Cultures were treated with 100 μM $H_2O_2$ at the time of measurement. Data are the mean ± SD. (*N* = 5 in E; *N* = 6 in F). **, **** *p* < 0.01 and *p* < 0.0001 as determined by one-way ANOVA (F) or *t* test (G). LB, Luria–Bertani; PBS, phosphate-buffered saline; WT, wild-type.

controls, *Salmonella* mutants bearing deletions in *greA* or *greB* genes were slightly attenuated in an acute C57BL/6 murine model of infection (**Fig 1A**). However, an isogenic strain carrying deletions in both *greA* and *greB* genes (Δ*greAB*) was highly attenuated in C57BL/6 mice (**Fig 1A**). Virulence of Δ*greAB Salmonella* could be complemented with either *greA* or *greB* genes driven by their native promoters from the low copy plasmid pWSK29, demonstrating that both Gre factors play indispensable, but mostly overlapping functions in *Salmonella* pathogenesis. Phagocyte NADPH oxidase (NOX2) and inducible nitric oxide synthase (NOS2) flavohemoproteins are key host determinants in resistance to *Salmonella* infections [2]. To examine if

the regulatory activity of Gre factors fosters *Salmonella* pathogenesis by dampening the antimicrobial actions of NOX2 or NOS2, we used *cybb*[-/-] and *nos2*[-/-] mice deficient in the gp91*phox* subunit of NOX2 and NOS2, respectively. *Salmonella* bearing deletions in both *greA* and *greB* genes remained attenuated in *nos2*[-/-] mice, but recovered virulence in *cybb*[-/-] mice (**Fig 1B**). These findings suggest that Gre factors contribute to the resistance of *Salmonella* to the oxidative stress engendered in the innate host response. In line with this observation, Δ*greA* and Δ*greB Salmonella* were hypersusceptible to $H_2O_2$ killing in vitro (**Fig 1C**). Simultaneous elimination of *greA* and *greB* genes further sensitized *Salmonella* to $H_2O_2$ killing (**Fig 1C**). The addition of $H_2O_2$ to bacterial cultures exacerbated the intrinsic growth defects of Δ*greAB Salmonella* in E salts minimum medium containing glucose and citric acid as carbon sources (EG) (**Fig 1D and 1E**).

Given the hypersusceptibility of Δ*greAB Salmonella* to ROS, we evaluated the capacity of this mutant strain to metabolize $H_2O_2$. Log phase Δ*greAB Salmonella* accumulated lower concentrations of $H_2O_2$ than wild-type controls (**Fig 1F**). A Δ*nuo* Δ*ndh* strain lacking both NADH dehydrogenases synthesized trace amounts of $H_2O_2$, pointing to NADH dehydrogenases as the main source of endogenous ROS. Further contributing to the low accumulation of $H_2O_2$, Δ*greAB Salmonella* harbored significantly ($p < 0.0001$) more peroxidatic activity than wild-type bacteria (**Fig 1G**). Together, this research suggests that *Salmonella* have leveraged the regulatory activity of Gre factors to resist oxidative stress generated in the innate host response by a mechanism that is independent of the detoxification of $H_2O_2$ by peroxidases.

## Gre factors resolve transcriptional errors in metabolic genes following exposure of *Salmonella* to oxidative stress

Gre proteins function as transcriptional fidelity factors, decreasing the inherent error rate of RNA polymerase [29–31]. To probe whether the observed hypersusceptibility of Δ*greAB Salmonella* to oxidative stress could be explained by decrease in transcriptional fidelity, we compared the transcription error rates of wild-type and Δ*greAB Salmonella* in the presence and absence of $H_2O_2$. Under peroxide stress, wild-type *Salmonella* suffered significant ($p < 0.01$) increases in transcription error rates (**Fig 2A**). In agreement with prior studies in *E. coli* [30], the overall transcriptional fidelity of Δ*greAB Salmonella* was significantly ($p < 0.001$) decreased when compared to wild-type controls. The global transcription error rate of Δ*greAB Salmonella* was not significantly affected by $H_2O_2$ treatment and was not significantly different from the error rate of wild-type *Salmonella* treated with $H_2O_2$, indicating that the hypersusceptibility of Δ*greAB Salmonella* to $H_2O_2$ is not due to a further exacerbation of the already elevated error rate under basal growth conditions.

To further dissect any changes in transcriptional fidelity that might explain the phenotype of Δ*greAB Salmonella* under oxidative stress, we next quantified specific single-nucleotide substitution (SNS) types (**Fig 2B, Table A in S1 Table**). While several individual substitution error types were significantly increased in untreated Δ*greAB* compared to untreated wild-type *Salmonella*, there did not appear to be any SNS type that drove the overall increased error rate in Δ*greAB Salmonella*. The lack of substitution error type bias between untreated wild-type and Δ*greAB Salmonella* conflicts with prior work in Δ*greAB E. coli* that identified a strong G>A substitution bias [29,30]. Similar to the untreated results, the increased overall error rate between untreated and $H_2O_2$-treated wild-type *Salmonella* did not appear to be driven by any specific substitution error type. These observed differences in specific SNS type bias between studies could be attributed to different assay conditions, high-throughput sequencing methods, or genuine differences in the inherent substitution type biases between organisms [32]. Combined, these results suggest that there is no specific SNS type that is inherently driving the overall transcription error rates in wild-type compared to Δ*greAB Salmonella*.

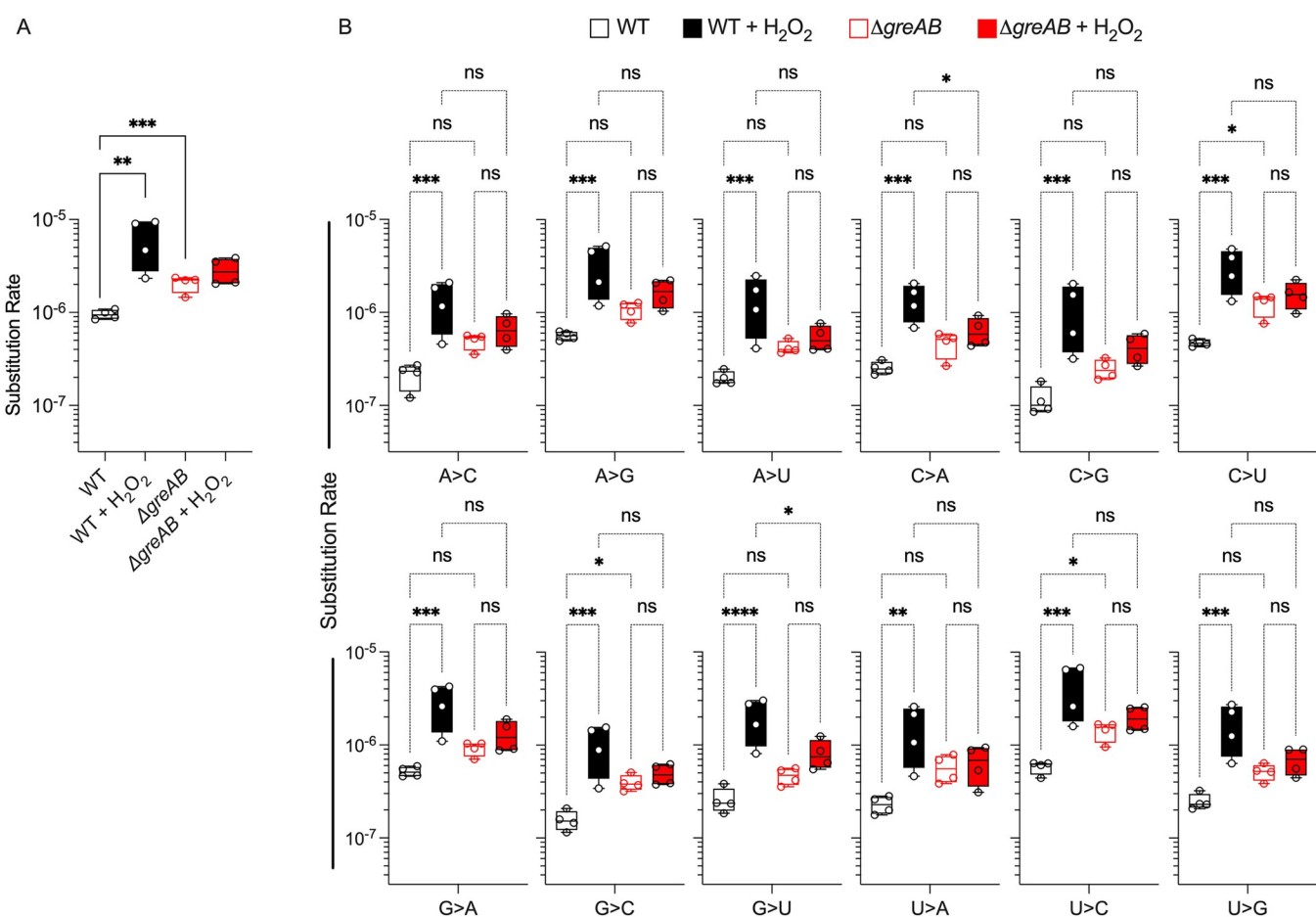

**Fig 2. Transcriptional fidelity in *Salmonella* undergoing peroxide stress.** (A) Quantification of overall transcription error rates in *Salmonella* grown in MOPS-GLC minimum medium to $OD_{600}$ of 0.25 in a shaking incubator. Selected samples were treated for 30 min with 400 μM $H_2O_2$. SNSs identified from RNA seq datasets were log-transformed prior to graphing. (B) Transcription error rates for specific nucleotide substitution types. Data were analyzed by unpaired Student's *t* test (A) two-way ANOVA (B). *, **, ***, ****, $p < 0.05$, $< 0.01$, $< 0.0001$, respectively. Box and whisker plots represent minimums to maximums, 25th and 75th percentiles, and medians (*N* = 4). SNS, single-nucleotide substitution; WT, wild-type.

To understand whether the differences in transcription error rate between wild-type and Δ*greAB Salmonella* were localized to specific transcripts or regions, enrichment analysis was performed. KEGG pathway enrichment analysis of transcripts harboring SNSs showed no significant and concordant differences in diverse metabolic pathways between wild-type and Δ*greAB* strains **(Fig a in S1 Text, Table B in S1 Table)**. However, enrichment analysis revealed that SNSs in transcripts associated with diverse metabolic pathways were no longer enriched upon oxidative stress **(Fig a in S1 Text, Table B in S1 Table)**. Conversely, in Δ*greAB Salmonella* SNSs in transcripts associated with diverse metabolic pathways were still enriched during oxidative stress. This observation suggests that Gre factors resolve transcriptional errors in transcripts encoding metabolic functions in *Salmonella* experiencing oxidative stress.

## Gre factors activate transcription of central metabolic genes

Given the high frequency in transcriptional errors in genes associated with metabolism in Δ*greAB Salmonella* after $H_2O_2$ treatment, we examined in further detail whether Gre factors generally affect metabolic output. As mentioned above, Δ*greAB Salmonella* grew poorly in

glucose minimum medium (**Fig 1D and 1E**). An extended lag phase was also observed when Δ*greAB Salmonella* were grown in MOPS minimal medium supplemented with either fructose, pyruvate, succinate, fumarate, or malate (**Panels A–J Fig b in S1 Text**). An extended lag phase has been previously described for *Salmonella* growing in dicarboxylic acids like succinate [33]. Amino acid analysis revealed that the content of valine, phenolalanine, and tyrosine was diminished in Δ*greAB Salmonella* (Panels A and B **Fig c in S1 Text**), suggesting that in the absence of Gre factors *Salmonella* experience nutritional shortages. The high content of the nucleotide alarmone guanosine tetraphosphate is a further sign that the Δ*greAB* strain is suffering from nutritional stress (**Panels C and D Fig c in S1 Text**). Wild-type and Δ*greAB Salmonella* grew with similar kinetics in MOPS minimal medium containing either casamino acids or a combination of glucose with all 20 amino acids (**Panels L and M Fig b in S1 Text**). Together, these findings suggest that Gre factors regulate assimilation of a variety of glycolytic sugars as well as various carbon sources that enter the TCA, allowing for the balanced production of amino acids.

Because glycolysis plays a salient role in resistance of *Salmonella* to the antimicrobial activity of NOX2 [14], we sought to examine the underlying causes that prevent Δ*greAB Salmonella* from effectively utilizing glucose. To delve into the mechanisms underlying the poor growth of Δ*greAB Salmonella* on glucose as sole carbon source, we compared the transcriptional profiles of log phase wild-type and Δ*greAB Salmonella* grown in MOPS-GLC minimal medium. Principal component analysis of the resulting RNA seq data showed marked differences in the transcriptomes of wild-type and Δ*greAB Salmonella* (**Panel A Fig d in S1 Text**) (GEO#GSE203342). Differential expression analyses revealed changes in the transcription of 1,569 genes between Δ*greAB* and wild-type *Salmonella* (FDR-corrected $p < 0.05$, $\log2 \geq |1|$), of which 944 were underexpressed and 623 were overexpressed in Δ*greAB Salmonella* (**Panel B Fig d in S1 Text and Table A in S2 Table**). With the exception of the *sitABCD* operon located within the SPI-1 locus that was up-regulated in *Salmonella* bearing mutations in *gre* genes, SPI-1 genes were poorly expressed by the mutant [34] (**Fig 3A and Table A in S2 Table**). Loci encoding peptidoglycan and lipopolysaccharide biosynthetic products were consistently down-regulated in Δ*greAB Salmonella* (**Fig 3A and Table A in S2 Table**), perhaps contributing to the defective growth of this mutant in MOPS-GLC medium (**Fig 1D and 1E and Panels A and K Fig b in S1 Text**). The down-regulation of SPI-2 genes in Δ*greAB Salmonella* (**Fig 3A and Table A in S2 Table**) may also greatly impact the virulence of this enteric pathogen.

ClueGO analysis of the differentially expressed genes indicated that Δ*greAB Salmonella* down-regulate oxidative phosphorylation (**Fig 3C**). In fact, gene clusters encoding the NADH dehydrogenase NDH-I and ATP synthase were expressed at lower levels in the mutant compared to wild-type controls (**Fig 3A**). In contrast, PTS system- and phosphate transport-encoding genes were up-regulated (**Fig 3A**). RT-PCR analysis confirmed the defective expression of *gapA*, *eno*, *nuoA*, *cydA*, and *atpE* genes encoding glyceraldehyde dehydrogenase (GAPDH), enolase, NADH dehydrogenase, cytochrome *bd*, and ATP synthase, respectively, in Δ*greAB Salmonella* (**Fig 3D**). The Δ*greAB Salmonella* strain may resolve the reduced carbon flow through lower glycolysis by up-regulating transcription of the *talA*-encoded aldolase, an enzyme that transfers 3 carbons from sedoheptulose 7-phosphate to glyceraldehyde 3-phosphate to form fructose 6-phosphate and the pentose phosphate pathway metabolite erythrose 4-phosphate. The Δ*greAB Salmonella* strain supported normal or increased expression of oxidative TCA genes involved in α-ketoglutarate synthesis, which may explain the accumulation of abnormally high concentrations of glutamic acid, citrulline, and ornithine in Δ*greAB Salmonella* grown on glucose (**Panel A Fig c in S1 Text**). The dysregulated expression of glycolytic and ETC genes may partially explain the poor utilization of glucose by Δ*greAB Salmonella*.

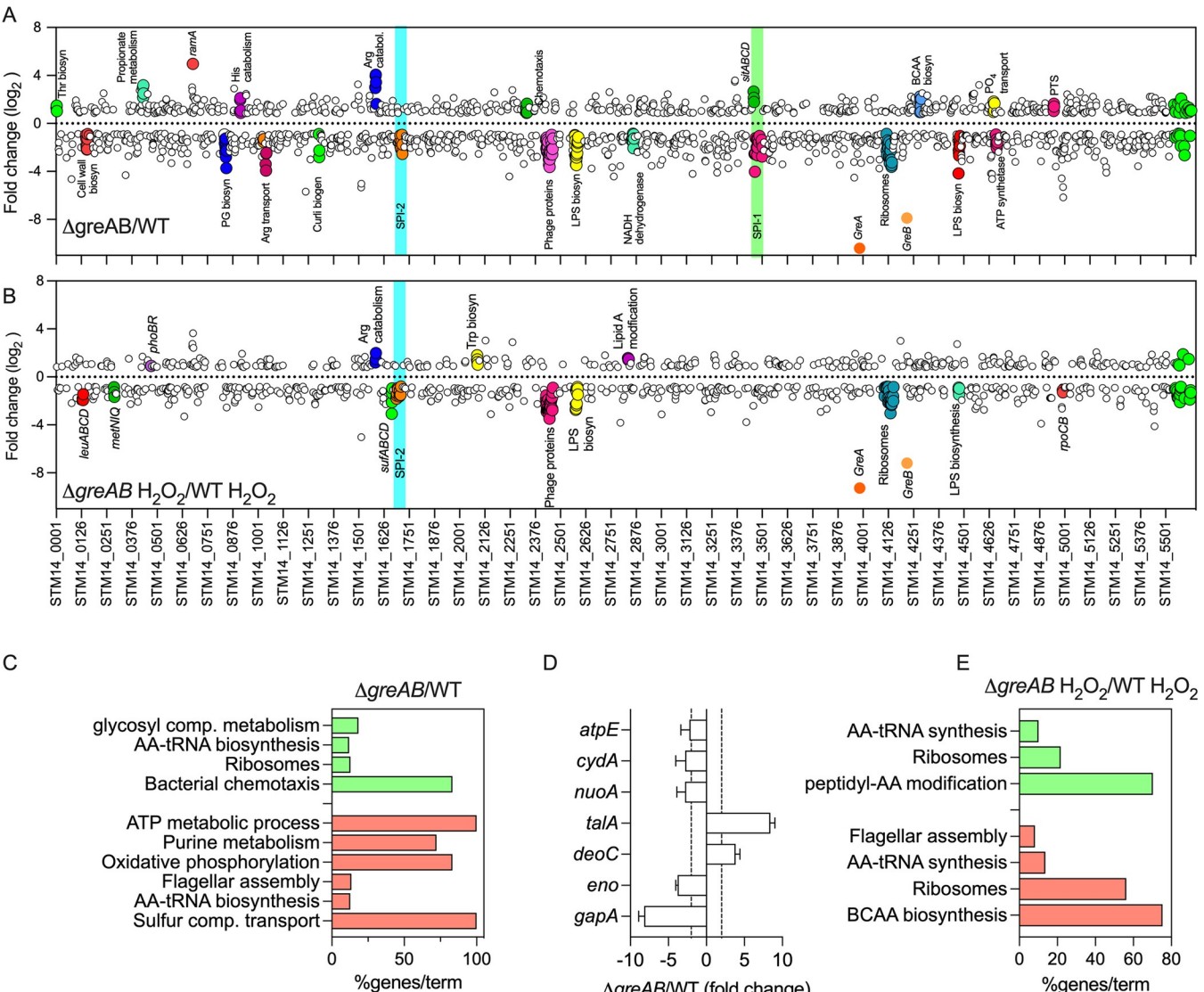

**Fig 3. Effect of Gre factors on the transcriptome of *Salmonella* sustaining oxidative stress.** Fold change values of gene expression in Δ*greAB Salmonella* compared to WT controls grown in MOPS-GLC medium (pH 7.2), in the absence (A) or presence (B) of 400 μM $H_2O_2$ for 30 min were determined by RNA-seq. Differentially expressed genes were mapped to the location in the chromosome (x axis). Color filled circles represent genetic operons of interest. Genes with a $\log_2$ fold change > 0.8 or < -0.8 are depicted on the top and bottom, respectively. Cyan and green boxes represent pathogenicity islands. (C, E) Gene enrichment analysis of differentially expressed genes in A and B was performed by the ClueGO app on cytoscape. Green and red colors represent up-regulated and down-regulated pathways, respectively. (D) Gene expression analysis by qRT-PCR of specimens isolated from WT or Δ*greAB Salmonella* grown to an $OD_{600}$ of 0.25 in MOPS-GLC minimal medium. The data, normalized to the *rpoD* housekeeping gene, represent the mean ± SD (*N* = 3). Dashed lines depict the 2-fold up- and down-regulated marks. qRT-PCR, quantitative real-time PCR; WT, wild-type.

We also compared the transcriptome of wild-type and Δ*greAB Salmonella* grown in MOPS-GLC medium following $H_2O_2$ treatment. Compared to untreated controls, principal component analysis showed a converging trend in the transcriptional profiles of wild-type and Δ*greAB Salmonella* after $H_2O_2$ treatment (**Panel A Fig d in S1 Text**) (GEO#GSE203342). Loci such as *katG*, *gshA*, *trxA*, *sodC-II*, *rpoS*, *dps*, *ftnA*, and *ftnB* encoding various antioxidant defenses were expressed to similar levels in wild-type and Δ*greAB Salmonella* following $H_2O_2$ treatment (**Table B in S2 Table**). On the other hand, the *cydA* gene, the *sufABCD* operon, and various SPI-2 genes were down-regulated in $H_2O_2$-treated Δ*greAB Salmonella* compared to

wild-type controls (**Figs 3B and Panel C Fig d in S1 Text and Table B in S2 Table**). Gene products involved in leucine, methionine, LPS, and ribosome biosynthesis were also repressed in ΔgreAB Salmonella in response to $H_2O_2$ compared to wild-type controls (**Fig 3B and 3E**). Together, these findings indicate that Salmonella do not seem to rely on the regulatory activity of Gre factors to activate transcription of key determinants associated with detoxification or scavenging of ROS. However, Gre proteins appear to be necessary for maximal activation of central metabolic genes associated with resistance to oxidative stress.

## Gre factors relieve transcriptional pauses in glycolytic genes

Given the essentiality of glycolysis in the resistance of Salmonella to oxidative killing [14], we next examined the mechanism by which Gre factors control transcription of glycolytic genes. The addition of GreA or GreB recombinant proteins (**Panel A Fig e in S1 Text**) to a reconstituted in vitro system increased expression of gapA (**Fig 4A**), a gene encoding the first enzyme in the payoff phase of glycolysis. To further probe the mechanism by which Gre factors directly promote gapA gene expression, we visualized the products of the in vitro transcription assays on urea PAGE gels. RNA polymerase paused at several sites between the +15 and +26 positions from the gapA transcription start site (**Fig 4B**). Addition of Gre factors, especially GreB, to the in vitro transcription reactions resolved the transcriptional pauses in the gapA gene. Gre factors also resolved transcriptional pauses in the eno gene encoding enolase (**Figs 4C and Panel B in Fig e in S1 Text**). DksA, which also binds to the secondary channel of RNA polymerase, did not resolve the transcriptional pauses occurring in the eno gene (**Panel B in Fig e in S1 Text**). Both Gre proteins increased gapA and eno transcriptional runoff products (**Fig 4B and 4C**), the presence of which is indicative of productive transcription elongation. In contrast to the glycolytic genes tested above, GreA and GreB proteins did not resolve pausing in dksA transcripts (**Fig 4D**), a gene that was not differentially expressed between ΔgreAB and wild-type Salmonella (**Table A in S2 Table**). Together, these investigations suggest that the Gre-dependent rescue of transcriptional pauses is an important step in the activation of key glycolytic genes in Salmonella.

## Glycolytic metabolism in ΔgreAB Salmonella undergoing oxidative stress

Given the regulation exerted by Gre factors on the transcription of glycolytic genes, we examined various glycolytic outputs in ΔgreAB Salmonella and wild-type controls. Wild-type Salmonella undergoing oxidative stress supported enhanced glucose uptake (**Fig 4E**), which is consistent with the idea that Salmonella undergo a glycolytic switch in response to oxidative stress [14]. $H_2O_2$-treated ΔgreAB Salmonella accumulated greater concentrations of glucose than wild-type controls (**Fig 4E**). Consistent with the transcriptional analysis, ΔgreAB Salmonella harbored less ($p < 0.05$) GAPDH enzymatic activity than wild-type controls grown in MOPS-GLC minimal medium (**Fig 4F**). Wild-type bacteria maintained excellent GAPDH activity upon $H_2O_2$ treatment (**Fig 4F**). In sharp contrast, the already diminished GAPDH activity in ΔgreAB Salmonella was highly susceptible to the inhibitory effects of $H_2O_2$ (**Fig 4F**). Interestingly, ΔgreAB Salmonella contained higher concentrations of 2-phosphoglycerate and pyruvate (**Fig 4G and 4H**) than wild-type Salmonella. These increased levels of the latter glycolytic intermediates in ΔgreAB Salmonella, which harbor low levels of GAPDH activity, may reflect the up-regulated transcription of Entner–Douderoff pathway genes such as dgaF (**Table A in S2 Table**), which shuttles carbon from the pentose phosphate pathway to lower glycolysis. Moreover, the high NADPH/NADP+ ratio in the mutant (**Fig 4I**) suggests that in the absence of Gre factors Salmonella directs a sizable fraction of the carbon entering glycolysis into the pentose phosphate pathway.

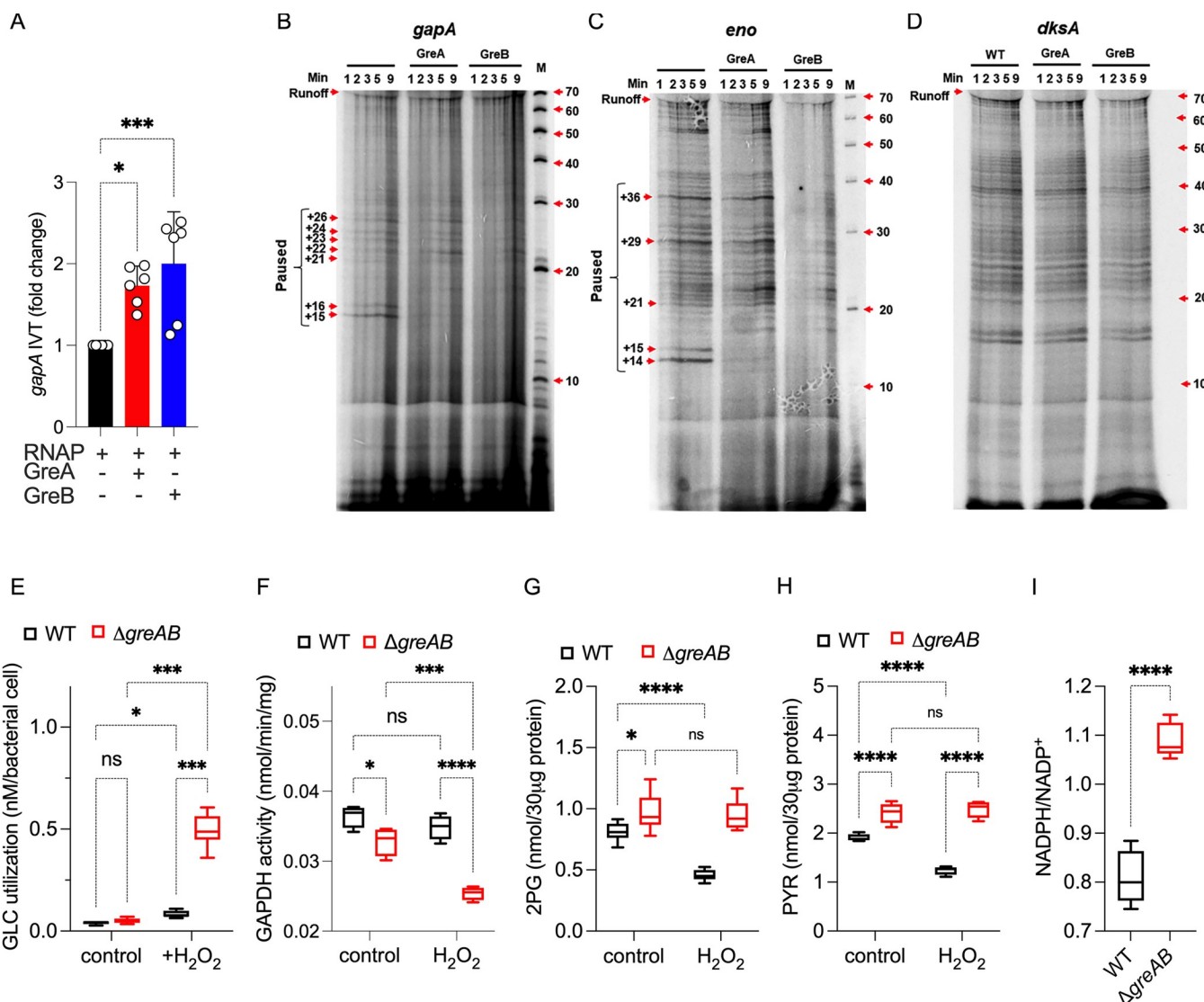

**Fig 4. Regulation of glycolytic transcription by Gre factors.** Effect of recombinant Gre proteins in the in vitro transcription of the *gapA* (A) gene in a reconstituted biochemical system. Recombinant GreA and GreB proteins were added at a final concentration of 150 nM and 50 nM, respectively. Transcription was measured by qRT-PCR. Data are shown as mean ± SD ($N = 6$). *, *** $p < 0.05$ and $p < 0.001$, respectively, as determined by one-way ANOVA. (B–D) Pausing of in vitro transcription reactions containing *gapA*, *eno*, or *dksA* templates was visualized in 7M urea-16% PAGE gels of RNA products labeled with $\alpha^{32}P$-UTP. The size of transcriptional pause products was identified by using $^{32}P$-labeled Decade Markers System and visualized by the Typhoon PhosphorImager. Representative blots from 3 independent experiments. Intracellular concentrations of glucose (GLC) (E), 2-phosphoglycerate (2-PG) (G), pyruvate (PYR) (H), and reduced and oxidized nicotinamide adenine nucleotide (I) in *Salmonella* grown aerobically to an $OD_{600}$ of 0.25 in MOPS-GLC medium (pH 7.2) at 37°C. Where indicated, bacterial cultures were treated with 400 μM $H_2O_2$ for 30 min. (F) The GAPDH activity in similarly treated *Salmonella* cultures was also assessed; $N = 4$. Statistical differences were calculated by two-way ANOVA (E–H) or Student's *t* test (I). qRT-PCR, quantitative real-time PCR; WT, wild-type.

## Gre factors activate aerobic respiration

Our transcriptional analyses have identified a critical function for Gre factors in the expression of genes encoding oxidative phosphorylation functions (**Fig 3A and 3C**), including the *cydA* locus that encodes a subunit of cytochrome *bd*. Therefore, we tested if Gre factors could directly activate *cydA* transcription. A reconstituted in vitro transcription system showed activation of *cydA* transcription by GreA and GreB proteins (**Fig 5A**). The activation of *cydA*

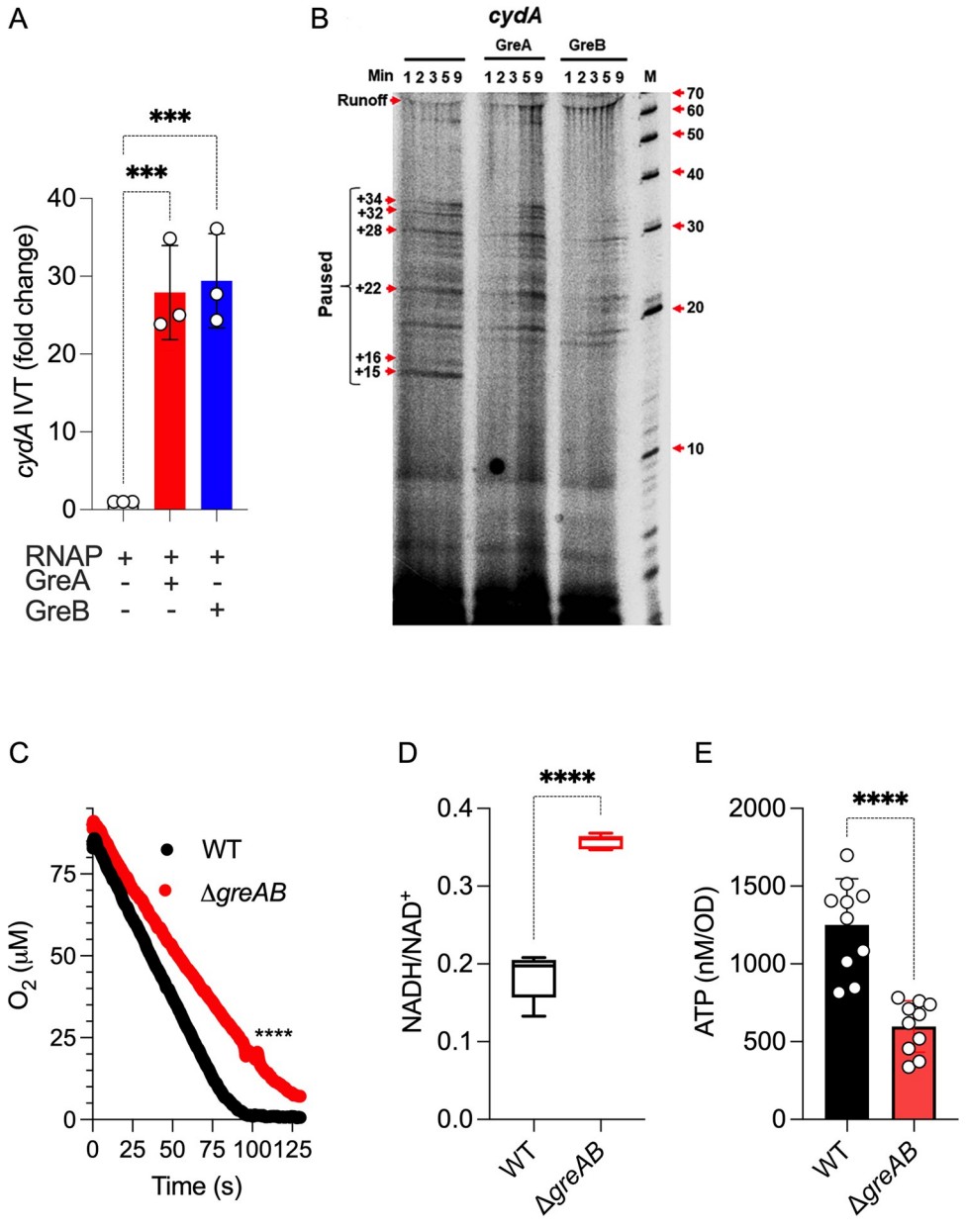

**Fig 5. Regulation of aerobic gene transcription by Gre factors.** (A) Effect of recombinant Gre proteins in the in vitro transcription of the *cydA* gene in a reconstituted biochemical system. Transcription was measured by qRT-PCR using conditions identical to the ones described in Fig 4. Data are the mean ± SD ($N = 6$). *** $p < 0.001$ as determined by one-way ANOVA. (B) Pausing of in vitro transcription reactions containing *cydA* templates was visualized in 7M urea-16% PAGE gels of RNA products labeled with $\alpha^{32}$P-UTP as described in Fig 4. Consumption of $O_2$ (C) by log phase *Salmonella* grown in MOPS-GLC medium (pH 7.2) was measured polarographycally in an APOLLO 4000 free radical anlyzer equipped with an ISO-OXY/HPO $O_2$ probe. Data are the mean ± SD ($N = 6$). **** $p < 0.0001$ as determined by unpaired *t* test. Reduced and oxidized nicotinamide adenine nucleotide (D) and intracellular ATP (E) were recorded in *Salmonella* grown aerobically to an OD$_{600}$ of 0.25 in MOPS-GLC medium (pH 7.2) at 37°C. Data are the mean ± SD (D, $N = 5$; E, $N = 10$). **** $p < 0.0001$ as determined by unpaired *t* test. qRT-PCR, quantitative real-time PCR; WT, wild-type.

transcription by Gre factors coincided with the resolution of transcriptional pauses (**Fig 5B**). In agreement with the transcriptional findings, Δ*greAB Salmonella* sustained lower aerobic respiration compared to wild-type bacteria (**Fig 5C**). Moreover, Δ*greAB Salmonella* harbored a

significantly ($p < 0.0001$) higher NADH/NAD$^+$ ratio and lower ATP concentrations than wild-type controls (**Fig 5D and 5E**), likely reflecting reduced transcription of NADH dehydrogenases and aerobic respiration genes (**Fig 3A and 3C**). Collectively, these investigations indicate that the transcriptional control Gre factors exert on ETC genes fosters aerobic metabolism, thereby helping *Salmonella* meet their energetic and redox needs.

### The ETC improves *Salmonella* growth on glucose and enhances resistance to oxidative stress

Our investigations have demonstrated that Gre factors facilitate transcription of the aerobic respiration gene *cydA* encoding a subunit of cytochrome *bd*. This terminal cytochrome imparts resistance to H$_2$O$_2$, NO, and hydrogen sulfide in diverse organisms [35–37]. The peroxidatic activity of cytochrome *bd* has been shown to protect *E. coli* from oxidative stress [38]. Moreover, the energetic output associated with cytochrome *bd* supports the pathogenesis of *Salmonella* [37]. Herein, we tested the importance of cytochrome *bd* in maintaining the energetics and fitness of *Salmonella* during periods of oxidative stress. A Δ*cydAB Salmonella* strain grew poorly in MOPS-GLC medium (**Fig 6A**). Moreover, mutations in the *atpB* gene encoding a subunit of ATP synthase, or in *nuo* and *ndh* genes encoding NDH-I and NDH-II NADH dehydrogenases also grew poorly on glucose medium (**Fig 6A**). In contrast to Δ*atpB* and Δ*nuo* Δ*ndh Salmonella*, Δ*cydAB* and Δ*greAB* strains grew as well as wild-type controls in MOPS-CAA minimum media (**Panel A in Fig f in S1 Text**). These findings are consistent with the idea that the energetic output-derived amino acid metabolism is highly dependent on the ETC [39]. Wild-type *Salmonella* contained a lower NADH/NAD$^+$ ratio than Δ*cydAB* controls (**Fig 6B**), consistent with the greater capacity of the former to perform aerobic respiration. The elevated NADH/NAD$^+$ ratios recorded in Δ*cydAB Salmonella* may contribute to the poor growth of this strain in glucose (**Fig 6A**), as NAD$^+$ is a cofactor of the glycolytic enzyme GAPDH. H$_2$O$_2$ treatment significantly ($p < 0.001$) elevated the NADH/NAD$^+$ ratio in both wild-type and Δ*cydAB Salmonella* (**Fig 6B**), potentially as a consequence of the oxidative damage of NDH-I NADH dehydrogenase [14]. The Δ*cydAB* strain also harbored reduced ATP content compared to wild-type controls (**Fig 6C**), likely reflecting the reduced aerobic respiration of the former. Both strains suffered reductions in ATP upon H$_2$O$_2$ treatment (**Fig 6C**), consistent with the down-regulation of aerobic respiration following exposure of *Salmonella* to oxidative stress [14]. Nonetheless, Δ*cydAB Salmonella* suffered significantly greater losses of ATP upon H$_2$O$_2$ treatment than wild-type controls (**Fig 6C**). At the culture density used in these experiments, 400 μM H$_2$O$_2$ was similarly bacteriostatic for both wild-type and Δ*cydAB Salmonella* (**Panel B in Fig f in S1 Text**), suggesting that the deficiencies in energetics recorded in Δ*cydAB Salmonella* are not likely explained by differences in bacterial growth. Collectively, these findings raise the possibility that the decreased expression of ETC genes like *cydAB* may contribute to the glycolytic defects of Δ*greAB Salmonella*.

H$_2$O$_2$ increased glucose utilization in both wild-type and Δ*cydAB Salmonella* (**Fig 6D**), consistent with the glycolytic switch favored in *Salmonella* undergoing oxidative stress. Despite increases in glucose uptake, wild-type and Δ*cydAB Salmonella* harbored lower concentrations of 2-phosphoglycerate and pyruvate after H$_2$O$_2$ treatment (**Fig 6E and 6F**). The decreased carbon flow through lower glycolysis may stem from the oxidation of the catalytic cysteine in GAPDH, favoring instead usage of the pentose phosphate pathway. In support of this idea, the NADPH/NADP$^+$ ratio was significantly ($p < 0.0001$) increased in wild-type *Salmonella* upon treatment with 400 μM H$_2$O$_2$ (**Fig 6G**). Interestingly, Δ*cydAB Salmonella* did not experience a similar increase in the NADPH/NADP$^+$ ratio after the addition of H$_2$O$_2$.

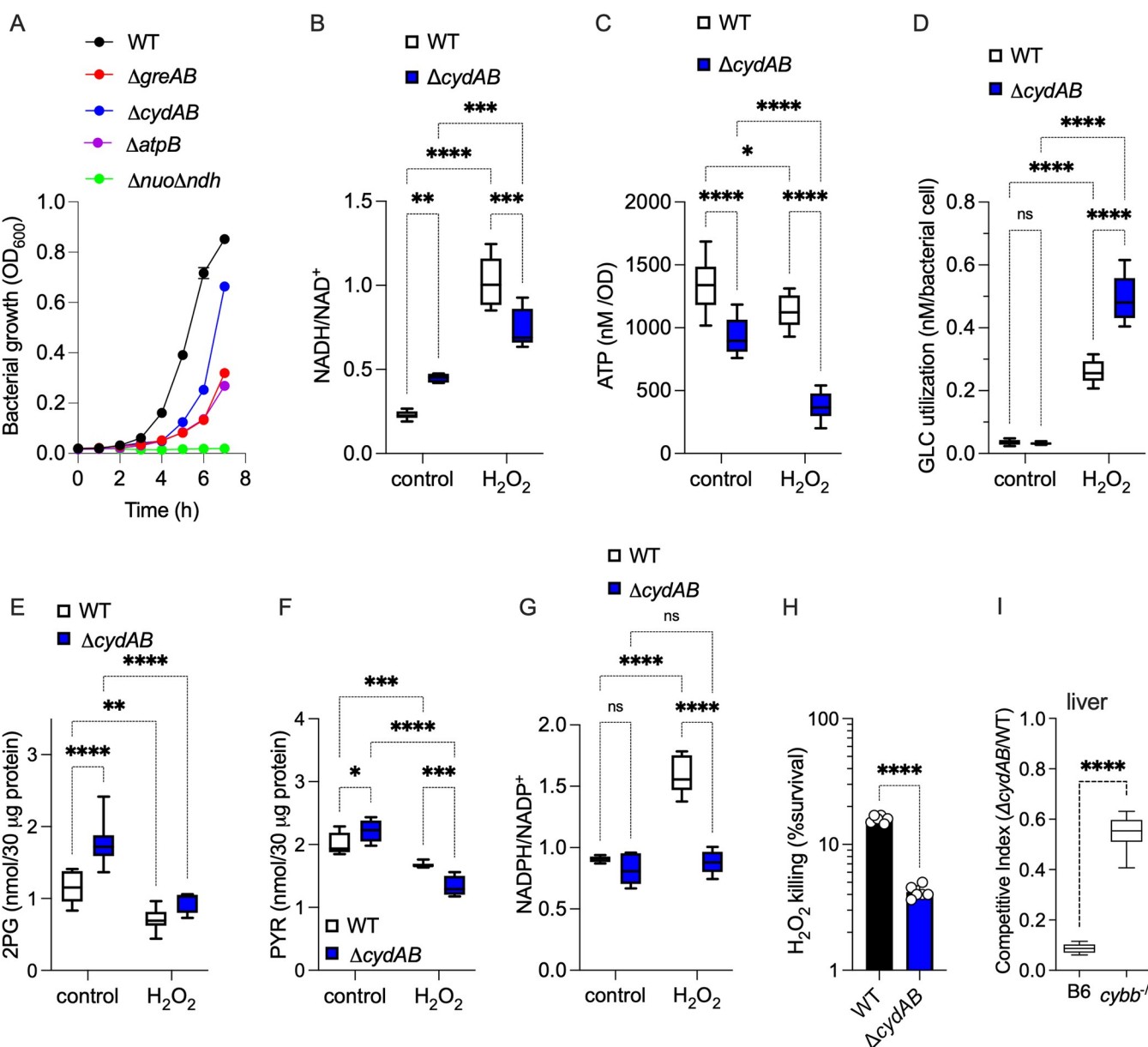

**Fig 6. Contribution of aerobic respiration to the antioxidant defenses of *Salmonella*.** (A) Aerobic growth of *Salmonella* strains in MOPS-GLC media (pH 7.2) at 37°C in a shaking incubator as assessed by $OD_{600}$. Data are the mean ± SD ($N$ = 3). Intracellular nicotinamide adenine nucleotide ratios (B, G), as well as the intracellular concentrations of ATP (C), glucose (D), 2-phosphoglycerate (2-PG) (E), and pyruvate (PYR) (F) in aerobic *Salmonella* grown to an $OD_{600}$ of 0.25 in MOPS-GLC medium (pH 7.2) at 37°C. Selected samples were treated with 400 μM $H_2O_2$ for 2 h (Panel D) or 30 min (Panels B, C, E, F, and G). Data are the mean ± SD ($N$ = 6–10). *, **, ***, ****, $p < 0.05$, $p < 0.01$, $p < 0.001$, and $p < 0.0001$, respectively, as determined by two-way ANOVA. (H) Bacterial cultures grown overnight in LB broth and diluted to $2 \times 10^5$ CFU/ml in PBS were treated for 2 h with 200 μM $H_2O_2$. Killing is expressed as percent survival compared to the bacterial burden at time zero. $N$ = 6; $p < 0.0001$ as determined by unpaired $t$ test. (I) Competitive index of *Salmonella* in livers of C57BL/6 and *cybb*[-/-] mice 3 days after i.p. inoculation with 100 CFU of equal numbers of WT and Δ*cydAB Salmonella* ($n$ = 10). ****, $p < 0.0001$ as determined by unpaired $t$ test. LB, Luria–Bertani; PBS, phosphate-buffered saline; WT, wild-type.

Having established important roles for cytochrome *bd* in energetics and glucose utilization, we proceeded to test whether this terminal cytochrome contributes to the resistance of *Salmonella* to oxidative stress. A *Salmonella* strain bearing mutations in the *cydAB* operon was hypersensitive to $H_2O_2$ killing (**Fig 6H**) and was attenuated in immunocompetent C57BL/6

mice (**Figs 6I and Panel C Fig f in S1 Text**). The attenuation of $\Delta cydAB$ *Salmonella* was partially reversed in $cybb^{-/-}$ mice (**Figs 6I and Panel C Fig f in S1 Text**), demonstrating that aerobic respiration is a previously unappreciated aspect that mediates resistance of *Salmonella* to ROS produced in the innate response. Cytochrome *bd* has peroxidatic activity [35], raising the possibility that products of the *cydAB* operon could protect *Salmonella* from oxidative stress by facilitating the direct detoxification of $H_2O_2$. However, wild-type and $\Delta cydAB$ *Salmonella* detoxified $H_2O_2$ with apparently similar ($p > 0.05$) kinetics (**Panel D Fig f in S1 Text**).

Cumulatively, these investigations suggest that aerobic respiration allows for optimal utilization of glycolysis and the pentose phosphate pathway, thus contributing to the antioxidant defenses that protect *Salmonella* against NOX2-mediated host immunity. It follows from this conclusion that the lower aerobic capacity recorded in $\Delta greAB$ *Salmonella* may contribute to the hypersusceptibility of this strain to ROS.

## Discussion

Our investigations demonstrate that the transcriptional regulation mediated by Gre factors promotes the resistance of *Salmonella* to ROS generated in the innate host response. The regulatory activity that Gre factors exert on transcriptional elongation and fidelity enables *Salmonella* to balance the utilization of sugars between glycolysis and aerobic respiration. The resulting energetic and redox signatures have been associated with the increased fitness of *Salmonella* during periods of oxidative stress [14].

The transcriptome of $\Delta greAB$ *Salmonella* contained more single-nucleotide substitution errors than wild-type controls grown in glucose. The poor quality of the transcriptome may contribute to the decreased fitness of $\Delta greAB$ *Salmonella*. Not surprisingly, the numbers of errors in the transcriptome increased following exposure of wild-type *Salmonella* to oxidative stress. However, there were no differences in the proportion of transcriptional errors between wild-type and $\Delta greAB$ *Salmonella* after $H_2O_2$ treatment. Nonetheless, the proportion of errors in metabolic genes was overrepresented in $\Delta greAB$ *Salmonella* undergoing peroxide stress. The combined actions that Gre factors exert on the fidelity and elongation of genes encoding central metabolic functions may promote *Salmonella* virulence and resistance to oxidative stress.

Transcriptional pauses that arise from the misincorporation of nucleoside monophosphates into the growing elongation complex are resolved by the Gre-directed endonuclease activity of RNA polymerase [26,40,41]. In addition to preserving transcriptional fidelity, Gre-mediated reactivation of stalled ternary elongation complexes reduces conflicts with the replisome [42–44], likely supporting bacterial growth [44,45]. Our investigations provide a non-mutually exclusive alternative for the growth-promoting activity of Gre factors. Gre-mediated resolution of transcriptional pauses in genes within EMP glycolysis and ETC helps *Salmonella* effectively utilize glucose, thereby satisfying the biosynthetic, energetic, and redox demands of the cell. However, the adaptive utilization of the pentose phosphate and Entner–Doudoroff pathways in $\Delta greAB$ *Salmonella* incompletely fulfills energetic and biosynthetic demands. Despite favoring NADPH-generating reactions in the pentose phosphate pathway, the underutilization of both glycolysis and aerobic respiration in $\Delta greAB$ *Salmonella* results in energetic shortages and redox stress, likely contributing to the hypersusceptibility of this strain to $H_2O_2$.

Our work demonstrates that the transcriptional activity of Gre factors stimulates expression of both glycolytic and aerobic respiration genes. Transcriptional activation of glycolysis by Gre factors may facilitate the survival of *Salmonella* during exposure to ROS [14]. In addition, our research has identified aerobic respiration as a previously unsuspected component of the adaptive response that facilitates resistance of *Salmonella* to NOX2-mediated oxidative killing. The role of cytochrome *bd* to the antioxidant defenses of *Salmonella* appear to be independent of

the peroxidatic activity of this terminal quinol oxidase [46], but rather dependent on the balanced utilization of glucose in glycolysis and the ETC. Cytochrome *bd* supports optimal growth of *Salmonella* on glucose, while yielding excellent ATP and NADPH outputs during periods of oxidative stress. A buildup of NADPH could also power antioxidant defenses such as thioredoxin or glutathione reductases. By regulating the expression of terminal cytochromes and other complexes of the ETC, Gre factors may help *Salmonella* resist oxidative stress.

We were perplexed by the high NADH/NAD$^+$ and NADPH/NADP$^+$ reducing power, as well as by the excellent peroxidatic activity recorded in Δ*greAB Salmonella*. In addition, the absence of Gre factors decreased aerobic respiration, thereby diminishing endogenous production of ROS. Despite these excellent predictors of bacterial resistance to oxidative stress, Δ*greAB Salmonella* lack the tolerance of wild-type cells to the bacteriostatic and bactericidal activity of ROS and are attenuated in NOX2 proficient mice. Thus, resistance to oxidative stress cannot be met with overflow metabolism alone, but requires the additional redox balancing and energetic outputs associated with aerobic respiration. Our research suggests that the metabolic adaptations that follow the resolution of transcriptional pauses at EMP and ETC genes is a *sine qua non* for resistance of *Salmonella* to oxidative stress. Moreover, antioxidant defenses such as peroxidases are not enough to overcome the susceptibility of bacteria to ROS without appropriate metabolic signatures that foster the balanced apportioning of resources to biosynthesis and redox production.

The partial suppression of oxidative phosphorylation in bacteria undergoing oxidative stress creates an energetic and redox balancing dilemma. *Salmonella* resolve this conundrum by increasing glucose utilization, which generates ATP in both lower glycolysis and through the fermentation of pyruvate to acetate. Increased glycolytic activity also boosts NADPH synthesis in the pentose phosphate pathway, fueling antioxidant defenses. This glycolytic switch also supplements the shrinking redox balancing capacity of an oxidatively damaged ETC by redirecting carbon to lactate fermentation and the reductive branch of TCA [14]. Our investigations indicate that *Salmonella* deficient in Gre factors do undergo this glycolytic switch in response to oxidative stress. Although Δ*greAB Salmonella* up-regulate glucose utilization upon oxidative stress, they lost much of their GAPDH activity following exposure to H$_2$O$_2$. Reduced GAPDH activity in Δ*greAB Salmonella* likely limits carbon utilization through phosphoglycerate mutase GpmA, a glycolytic enzyme that is associated with the protective response of *Salmonella* and *E. coli* against oxidative stress [14,47].

Our research has shown that Δ*greAB Salmonella* expressed normal levels of *katG*, *trxA*, or *sodC* genes, and exhibited excellent peroxidatic activity, suggesting that the poor tolerance exhibited by this mutant to peroxide stress may not be explained by defects on antioxidant defenses that rely on the detoxification of ROS. The selective expression of these antioxidant determinants following peroxide stress is likely dependent on the regulatory activity of transcription factors such as SoxR, OxyR, PhoP, RpoS, or RpoE. On the other hand, the metabolic defects arising from deletion of *gre* genes likely predispose *Salmonella* to oxidative killing. Having said that, we would like to point out that the effects on energetics and redox balancing stemming from the regulation of transcription elongation of central metabolic genes are probably only one of mechanisms by which Gre factors promote pathogenesis and resistance of *Salmonella* to oxidative stress. In addition to activating transcription of genes encoding glycolytic and ETC functions, Gre factors promote transcription of branch chain amino acids and methionine biosynthesis genes, the *sufABCD* operon and the SPI-2 locus, all of which are vital determinants associated with bacterial resistance to reactive species [6,48–53]. Future investigations will be needed to establish if the expression of *leuABCD*, *metNIQ*, *sufABCD*, and SPI-2 genes in *Salmonella* undergoing peroxide stress reflects direct transcriptional regulation by Gre factors.

Most studies have focused on the regulation of metabolism that follows the hierarchical control provided by transcriptional activators as well as the allosteric regulation of central metabolic enzymes by metabolites and posttranslational modifications. In complement to these studies, our investigations demonstrate that both the resolution of transcriptional pauses and maintenance of transcriptional fidelity are key for metabolic outputs associated with resistance to oxidative stress. Gre-dependent regulation of transcription of EMP and ETC genes balances the simultaneous usage of overflow metabolism and aerobic respiration, thus fulfilling the biosynthetic, energetic, and redox requirements that help *Salmonella* withstand the antimicrobial activity of NOX2 during the acute host response. In addition to the effects in metabolism characterized in our investigations, the global effects Gre factors have on the transcriptome likely add in as yet unsuspected ways to the resistance to oxidative stress and bacterial pathogenesis.

## Materials and methods

### Bacterial strains, plasmids, and growth conditions

The *Escherichia coli* strains DH5α and BL21(DE3)pLysS were grown in Luria–Bertani (LB) broth or agar at 37°C. *Salmonella enterica* serovar *Typhimurium* strain 14028s (ATCC, Manassas, Virginia, United States of America) and its mutant derivatives were grown in LB broth or E-salts minimal medium [57.4 mM $K_2HPO_4$, 1.7 mM $MgSO_4$, 9.5 mM citric acid, and 16.7 mM $H_5NNaPO_4$ (pH 7.0), supplemented with 0.1% casamino acids, and 0.4% D-glucose (EGCA)], or MOPS minimal medium [40 mM MOPS buffer, 4 mM Tricine, 2 mM $K_2HPO_4$, 10 μM $FeSO_4 \cdot 7H_2O$, 9.5 mM $NH_4Cl$, 276 μM $K_2SO_4$, 500 nM $CaCl_2$, 50 mM NaCl, 525 μM $MgCl_2$, 2.9 nM $(NH_4)_6Mo_7O_{24} \cdot 4H_2O$, 400 nM $H_3BO_3$, 30 nM $CoCl_2$, 9.6 nM $CuSO_4$, 80.8 nM $MnCl_2$, and 9.74 nM $ZnSO_4$ (pH 7.2)] supplemented with 0.4% D-glucose or 0.4% casamino acids obtained by acid hydrolysis of casein at 37°C in a shaking incubator. Ampicillin (100 μg/ml), kanamycin (50 μg/ml), chloramphenicol (20 μg/ml), and tetracycline (20 μg/ml) were used where appropriate. **Tables a and b in S1 Text** list the strains and plasmids used in this study.

### Construction of *Salmonella* Δ*greAB* mutants and complementation

Deletion mutants were constructed using the λ-Red homologous recombination system [54]. Specifically, the chloramphenicol cassette from the pKD3 plasmid and kanamycin cassette from pKD13 were PCR amplified using primers with a 5′-end overhang homologous to the bases following the ATG start site and the bases preceding the stop codon of *greA* and *greB* genes, respectively (**Table c in S1 Text**). The PCR products were gel purified and electroporated into *Salmonella* expressing the λ Red recombinase from the plasmid pTP233. Transformants were selected on LB plates containing 10 μg/ml chloramphenicol or 50 μg/ml kanamycin. To construct a mutant deficient in both Gre factors, the Δ*greB*-Km mutation was moved into the Δ*greA*-Cm mutant via P22-mediated transduction, and the pseudolysogens were eliminated by streaking on Evans blue uridine agar plates. Transformants were selected on 50 μg/ml kanamycin and 20 μg/ml chloramphenicol LB agar plates. The mutants were confirmed by PCR and sequencing.

The Δ*greAB* mutant was complemented with *greA* or *greB* genes expressed from the low-copy pWSK29 plasmid [55]. The *greA* and *greB* coding regions plus a 400 bp upstream region including the native promoter were PCR amplified using *greA* pro F and *greA* R or *greB* pro F and *greB* R primers, respectively (**Table c in S1 Text**). The amplified PCR products were directly cloned into the SacII and BamHI restriction sites at the MCS of the pWSK29 vector. The resulting Δ*greAB*::*greA* or Δ*greAB*::*greB* complemented strains were selected on 250 μg/ml penicillin LB agar plates.

## Cloning, expression, and purification of proteins

Recombinant 6XHis-tag GreA or 6XHis-tag GreB were produced by cloning *greA* and *greB* genes into NdeI and BamHI sites of the pET14B vector (Novagen) using *greA* F and *greA* R or *greB* F and *greB* R primers, respectively **(Tables b and c in S1 Text).** All constructs were confirmed by sequencing. Plasmids were expressed in *E. coli* BL21 (DE3) pLysS (Invitrogen). Cells grown in LB broth at 37°C to an $OD_{600}$ of 0.5 were treated with 1 mM isopropyl β-D-1-thiogalactopyranoside. After 3 h, the cells were harvested, disrupted by sonication, and centrifuged to obtain cell-free supernatants. 6XHis-tag fusion proteins were purified using Ni-NTA affinity chromatography (Qiagen) as per manufacturer's instructions. DksA protein was purified as described previously [56]. A GST-DksA fusion protein was purified using Glutathione-Sepharose 4B (bioWORLD, Dublin, Ohio, USA) according to manufacturer's protocols. To remove the GST tag, PreScission protease was added to recombinant GST-DksA protein in phosphate-buffered saline (PBS) containing 10 mM DTT. After overnight incubation at 4°C, proteins were eluted and further purified by size-exclusion chromatography on Superdex 75 (GE Healthcare Life Sciences). Purified DksA proteins were aliquoted inside a BACTRON anaerobic chamber (Shel Lab, Cornelius, Oregon, USA). The purity and mass of the recombinant proteins were assessed by SDS/PAGE.

## Ethics statement

This study was performed in accordance with the recommendations in the Guide for the Care and Use of Laboratory Animals of the National Institutes of Health. All animals were handled in accordance with the Guide for the Care and Use of Laboratory Animals, following the approved Institutional Animal Care and Use Committee (IACUC) protocol 00059 of the University of Colorado School of Medicine (Assurance Number A3269-01), an AAALAC Accredited Institution.

## Animal studies

Six to 8-week-old immunocompetent C57BL/6, and immunodeficient nos2$^{-/-}$ or *cybb*$^{-/-}$ mice deficient in the inducible nitric oxide synthetase or the gp91$^{phox}$ subunit of the NADPH oxidase, respectively, were inoculated intraperitoneally with approximately 100 CFU of *Salmonella* grown overnight in LB broth at 37°C in a shaking incubator. Mouse survival was monitored over 14 days. The bacterial burden was quantified in livers and spleens 3 days post infection by plating onto LB agar containing the appropriate antibiotics. Competitive index was calculated as (strain 1/strain 2)$_{output}$/(strain 1/strain 2)$_{input}$. The data are representative of 2 to 3 independent experiments. All mice experiments were conducted according to protocols approved by the Institutional Animal Care and Use Committee at the University of Colorado School of Medicine.

## Susceptibility to $H_2O_2$

*Salmonella* strains grown overnight in LB broth at 37°C with shaking were diluted in PBS to a final concentration of $2 \times 10^5$ CFU/ml. The cells were treated with 200 μM of $H_2O_2$ at 37°C for 2 h. The surviving bacteria were quantified after plating 10-fold serial dilutions onto LB agar. The percent survival was calculated by comparing the surviving bacteria after $H_2O_2$ challenge to the starting number of cells. The effect of $H_2O_2$ on bacterial growth was also examined. Briefly, *Salmonella* strains grown overnight in LB broth at 37°C with shaking were subcultured 1:100 into EG minimal medium, and 200 μl were seeded onto honeycomb microplates and

treated with or without 400 μM $H_2O_2$. The $OD_{600}$ was recorded every 15 min for up to 40 h in a Bioscreen C plate reader (Growth Curves USA).

## Growth kinetics

Overnight *Salmonella* cultures grown in MOPS-GLC medium with appropriate antibiotics were diluted 1:100 into fresh MOPS-GLC medium. Where indicated, 0.4% D-glucose, 40 μg/ml of each amino acid, 0.4% casamino acids, or 0.15% TCA intermediates was added to MOPS-GLC medium. In addition, MOPS minimal medium was supplemented with 0.4% maltose, fructose, galactose, or lactose as needed. *Salmonella* were also grown in E-salts minimal medium supplemented with 0.4% D-glucose (EG). Bacterial growth was followed by recording $OD_{600}$ values every hour for 7 to 10 h at 37°C in an aerobic shaking incubator or anaerobic chamber.

## Thin layer chromatography

Nucleotides were examined as originally described with minor modifications [50,57]. Briefly, *Salmonella* strains grown overnight in MOPS-GLC medium supplemented with 2 mM $K_2HPO_4$ were diluted 1:100 into fresh 0.4 mM $K_2HPO_4$ MOPS-GLC medium. The cultures were grown to early exponential phase till the $OD_{600}$ reached approximately 0.2. One-milliliter culture aliquots were labeled with 10 μCi of inorganic $^{32}P$. After 1 h, the cells were treated with 0.4 ml of ice-cold 50% formic acid and incubated on ice for at least 20 min. The extracts were centrifuged at 16,000×g for 5 min. A 2.5- or 5-μl volume of ice-cold extracts were spotted along the bottom of polyethyleneimine-cellulose TLC plates (Millipore). The spots were air dried, and the TLC plates were placed into a chamber containing 0.9 M $K_2HPO_4$ (pH 3.4). $^{32}P$-labeled nucleotides in the TLC plates were visualized with phosphor screens on a phosphorimager (Bio-Rad), and relative nucleotide levels were quantified with the ImageJ software (NIH).

## Polarographic $O_2$ and $H_2O_2$ measurements

Consumption of $O_2$ was measured using an ISO-OXY-2 $O_2$ sensor attached to an APOLLO 4000 free radical analyzer (World Precision Instruments, Sarasota, Florida, USA) as described [58]. Briefly, 3 ml of *Salmonella* grown aerobically to $OD_{600}$ of 0.25 in MOPS-GLC medium were rapidly withdrawn, vortexed for 1 min and immediately recorded for $O_2$ consumption. A two-point calibration for 0% and 21% $O_2$ was done as per manufacturer's instructions. $H_2O_2$ was measured in an APOLLO 4000 free radical analyzer using an $H_2O_2$-specific probe. $H_2O_2$ production by Δ*nuo* Δ*ndh Salmonella* was measured after 12 h of growth, when the cultures reached an $OD_{600}$ of 0.25.

## NAD(P)H and NAD(P)$^+$ measurements

Intracellular NAD(P)H/NAD(P)$^+$ measurements were carried out according as described [58] with slight modifications. Briefly, *Salmonella* grown in MOPS-GLC medium at 37°C to an $OD_{600}$ of 0.25 were treated for 30 min with or without 400 μM $H_2O_2$. NAD(P)H and NAD(P)$^+$ were extracted from pellets in 0.2 M NaOH or 0.2 M HCl. Ten microliter of extracts were added to 90 μl of reaction buffer containing 200 mM bicine (pH 8.0), 8 mM EDTA, 3.2 mM phenazine methosulfate, and 0.84 mM 3-(4,5-dimethylthiazol- 2-yl)−2,5-diphenyltetrazolium bromide. NAD$^+$/NADH and NADP$^+$/NADPH concentrations were estimated in reactions containing 20% ethanol and 0.4 μg alcohol dehydrogenase or 2.54 mM glucose-6-phosphate and 0.4 μg glucose-6-phosphate dehydrogenase, respectively. NADH(P) and NAD(P)$^+$ measured at 570 nm for the thiazolyl tetrazolium blue cycling assay and calculated by regression analysis of known standards, and specimens were standardized according to $OD_{600}$.

## LC-MS amino acid analysis

To measure amino acids by LC-MS, approximately $5 \times 10^{10}$ Salmonella were collected from cultures grown in MOPS-GLC medium at 37˚C to an $OD_{600}$ of 0.25. Amino acids were extracted on ice-cold lysis buffer [5:3:2 ratio of methanol-acetonitrile-water (Thermo Fisher Scientific, Pittsburgh, Pennsylvania, USA)] containing 3 μM of amino acid standards [Cambridge Isotope Laboratories, Tewksbury, Massachusetts, USA]. Samples were vortexed for 30 min at 4˚C in the presence of 1-mm glass beads. Insoluble proteins and lipids were pelleted by centrifugation at 12,000 x g for 10 min at 4˚C. The supernatants were collected and dried with a SpeedVac concentrator. The pellets resuspended in 0.1% formic acid were analyzed in a Thermo Vanquish ultra-high-performance liquid chromatography (UHPLC) device coupled online to a Thermo Q Exactive mass spectrometer. The UHPLC-MS methods and data analysis approaches used were described previously [59].

## Glucose utilization

Glucose in the culture medium was measured by the Glucose Assay Kit (Abcam) as per manufacturer's instructions. Briefly, Salmonella grown in MOPS-GLC medium at 37˚C to $OD_{600}$ of 0.25 were harvested and resuspended in fresh MOPS-GLC media. Selected samples were treated with 400 μM $H_2O_2$ at 37˚C. Culture supernatants were collected at 2 h after treatment and stored at −20˚C until further use. Five microliter of culture supernatants mixed with 400 μl o-toluidine reagent were incubated at 100˚C for 8 min. Reaction mixtures were cooled down in ice for 5 min and the OD was recorded at 630 nm. Glucose concentration was calculated by regression analysis of known glucose standard.

## 2-phosphoglycerate and pyruvate estimation

2-phosphoglycerate (2-PG) and pyruvate were measured in Salmonella grown in MOPS-GLC medium at 37˚C to $OD_{600}$ of 0.25. Where indicated, cells were challenged for 30 min with 400 μM $H_2O_2$. Bacterial cells were harvested and sonicated in 200 μl of ice-cold lysis buffer (25 mM Tris-HCl (pH 8.0), 100 mM NaCl). Soluble cytoplasmic extracts obtained after clarification at 13,000 g for 10 min at 4˚C were processed with the 2-phosphoglycerate Assay Kit (Abcam) as per manufacturer's instructions. 2-PG and pyruvate concentrations in the lysates were calculated by linear regression using known 2-PG and pyruvate standards.

## GAPDH enzymatic activity

GAPDH activity in Salmonella was measured by the GAPDH activity assay kit (Abcam) as per manufacturer's instructions. Briefly, Salmonella were grown in MOPS-GLC medium at 37˚C to an $OD_{600}$ of 0.25. Cells were sonicated in 200 μl of ice-cold lysis buffer (25 mM Tris-HCl (pH 8.0),100 mM NaCl). Insoluble material was removed by centrifugation at 13,000 g for 10 min at 4˚C. GAPDH enzymatic activity in soluble cytoplasmic extracts was estimated by measuring the accumulation of NADH at 450 nm formed in conversion of glyceraldehyde-3-phosphate into 1, 3-bisphosphate glycerate. GAPDH activity was standardized to equal amounts of protein. GAPDH activity was calculated by linear regression using known NADH standard.

## ATP measurements

ATP concentrations were quantified with the luciferase-based ATP determination kit (Molecular Probes). Briefly, Salmonella grown in MOPS-GLC medium at 37˚C to an $OD_{600}$ of 0.25 were challenged for 30 min with and without 400 μM $H_2O_2$. Pellets from 2 ml cultures were thoroughly mixed with 0.5 ml of ice-cold, 380 mM formic acid containing 17 mM EDTA.

After centrifugation for 1 min at 16,000×g, supernatants were diluted 25-fold into 100 mM N-tris(hydroxymethyl)methyl-2-aminoethanesulfonic acid (TES) buffer (pH 7.4). Ten microliter of samples or ATP standards were mixed with 90 μl of reaction master mix (8.9 ml of water, 500 μl of 20× buffer, 500 μl of 10 mM D-luciferin, 100 μl of 100 mM dithiothreitol [DTT], 2.5 μl of 5-mg/ml firefly luciferase). Luminescence was recorded in an Infinite 200 PRO (Tecan Life Sciences). ATP concentrations were calculated by linear regression using ATP standards, and the intracellular concentration of ATP was standardized to CFU/ml.

## RNA isolation, library preparation, and RNA seq

*Salmonella* grown in MOPS-GLC medium at 37˚C to an $OD_{600}$ of 0.25 were treated with 1 ml of TRIzol reagent (Life Technologies). Following chloroform extraction, RNA was precipitated from the aqueous phase by the addition of 3 M sodium acetate (1/10, vol/vol), 50 mg/ml glycogen (1/50, vol/vol), and an equal volume of 100% isopropyl alcohol. Precipitated RNA was washed twice with 70% (vol/vol) ethanol, suspended in RNase free $dH_2O$, and treated with RNase free DNase I, according to the supplier's specifications (Promega). Reactions were terminated by the addition of an equal volume of phenol/chloroform/isoamyl alcohol solution (25:24:1) (PCI). The aqueous phase was treated with an equal volume of chloroform. RNA in the resulting aqueous phase was precipitated by the addition 3 M sodium acetate (1/10 vol/vol), 50 mg/ml glycogen (1/50 vol/vol), and 3 volumes of 100% ethanol. The quality of the isolated RNA was assessed on an Agilent Bioanalyzer. Ribosomal RNA was removed from the total RNA preparation using the MICROBExpress kit (Life Technologies). Starting with 1 μg purified mRNA, samples were fragmented with the NEB Magnesium Fragmentation module at 94˚C for 5 min. RNA was purified by PCI extraction and ethanol precipitation and sodium acetate, and libraries were prepared for Illumina sequencing by following the protocol accompanying the NEBNext Ultra RNA Library Prep Kit through completion of the second strand synthesis step. Libraries were made by NEBNext Ultra RNA Library Prep Kit protocol for a target insert size of 300 bp. Samples were barcoded using NEBNext Multiplex Oligos (Universal primer, Index Primers Set 1 and Index Primers Set 2), and the resulting indexed libraries were sequenced on an Illumina MiSeq using 300-nt reads. The i7 Illumina adapters were trimmed from raw paired reads by utilizing Cutadapt version 2.10 in the Linux terminal with the sequences AGATCG GAAGAGCACACGTCTGAACTCCAGTCAC and AGATCGGAAGAGCGTCGTGTAGG GAAAGAGTGTAGATCTCGGTGGTCGCCGTATCATT for the forward and reverse reads, respectively. Reads were then mapped with Bowtie2 [60,61] version 2.3.2 using CP001363.1 and CP001362.1 [62] as the reference genome for *S. Typhimurium 14028s*. Picard version 2.18.27 was then used to remove duplicates and sort the reads. HTseq [63] version 0.13.5 was then leveraged to generate count files by locus for each sample. Counts for each sample were then statistically analyzed utilizing DEseq2 1.30.1 [64] and edgeR 3.32.1 [65,66] in R Studio running R version 4.0.4 by using Fisher's exact test on the tagwise dispersion of counts for loci that had at least 80 reads total across all samples be analyzed. Genes categorized following KEGG annotations were imported with heatmap 1.0.12 in R for graphical representation along the following color breaks for fold-change values of: 0.1386, 0.6060, 1.0000, 2.1304, 2.8125, and 7.5938. Volcano plots were generated with EnhancedVolcano in R. PCA analysis was performed after a log transformation and Pareto scale of the raw counts data. Final heatmaps, PCA, and loadings graphs were manipulated in Inkscape version 0.92.1 to add labels and overlay findings.

## Transcriptional fidelity data analysis

I7 Illumina adaptors were trimmed from raw paired reads with Trim Galore (version 0.6.7) using the first 13 bp of the standard Illumina adaptor sequence, AGATCGGAAGAGC. Reads

were mapped with Bowtie2 (version 2.5.0) using the *S. Typhimurium* 14028s primary genome assembly GCA_000022165.1, which contains both CP001363.1 and CP001362.1, as the reference genome. Samtools (version 1.6) was then used to sort the alignments and remove singletons [67]. The Bayesian genetic variant detector, freebayes (version 1.0.2), was used to call variants on the alignments using stringent quality cutoffs (-m 30, -q 20) to minimize variant miscalls [68]. Variants were annotated with bcftools [67] (version 1.15.1) and a custom feature table curated from the NCBI website. Annotated variants were then subjected to call quality filters, and SNSs were selected using vcflib (version 1.0.0_rc2) [69]. Total reads and specific nucleotides sequenced were determined with Pysamstats (version 1.1.2) using the same stringent mapping and base quality cutoffs that were used for variant calling (—min-map1 = 30,—min-baseq = 20). Known multicopy genes (tuf_1, tuf_2, 23S, 16S, 5S rRNA) were removed from the datasets to minimize potential sources of error that could arise from ambiguous read mapping. To facilitate identification of RNA polymerase-specific errors, SNSs not within the annotated transcriptome and SNSs with >2 alternate allele observations were filtered from the dataset, and complementary mismatch pairs were used for variants identified on the antisense strand. Total error rates were normalized on a per sample basis by dividing the total number of unique SNSs identified by the total number of bases sequenced by. Nucleotide substitution error rates were also normalized on a per sample basis by dividing the total number of each substitution type identified by the total number of each reference base sequenced. Any data filtering and calculations performed outside of the listed packages was accomplished with custom R (version 4.2.1) scripts run in RStudio.

KEGG pathway overrepresentation analysis of transcripts containing SNSs was performed using clusterProfiler (version 4.6.0) [70,71], and enrichplot (version 1.18.3) was used for graphical representation of significantly enriched (adjusted *p*-value of ≤0.05) pathway terms. To visualize the distribution of transcription errors across the transcriptome, the number of unique SNSs per transcript was quantified and then mapped onto the full transcriptome of *S. Typhimurium* 14028s. The resulting transcriptional profile of SNSs was imported into Prism (version 9.4.1) for visualization.

### RNA isolation and quantitative RT-PCR

*Salmonella* strains grown in MOPS-GLC medium in a shaking incubator at 37˚C to an $OD_{600}$ of 0.25 were centrifuged at 16,000×g for 10 min at 4˚C. The bacterial pellets were saved at −80˚C until further processing. DNA-free RNA was purified using a High Pure RNA isolation kit (Roche) according to the manufacturer's instructions. First-strand cDNA generation from total RNA was generated using Moloney murine leukemia virus (M-MLV) reverse transcriptase (Promega). Relative mRNA quantitation was done using the SYBR green quantitative real-time PCR (qRT-PCR) master mix (Roche) using the primers described in **Table c in S1 Text**. Data evaluation of 3 biological replicates done in triplicate was performed using the threshold cycle ($2^{-\Delta\Delta CT}$) method. Gene expression was normalized to internal levels of the housekeeping gene *rpoD*. Transcripts that exhibited 2-fold up- or down-regulation were considered to exhibit a significant change.

### In vitro transcription

Products synthesized in in vitro transcription reactions were quantified by qRT-PCR as described previously [23,56]. Transcription reactions were performed in 40 mM HEPES (pH 7.4), 2 mM $MgCl_2$, 60 mM potassium glutamate, 0.1% Nonidet P-40, 200 μM of each ATP, GTP, CTP, and UTP (Thermo Fisher Scientific, Grand Island, New York, USA), 8 U RiboLock RNase inhibitor (Thermo Fisher Scientific, Grand Island, New York, USA), 1 nM of DNA

template, 5 nM *E. coli* holoenzyme RNA polymerase (New England Biolabs, Ipswich, Massachusetts, USA). Where indicated, 150 nM of GreA or 50 nM of GreB proteins were added to the in vitro transcription reactions. Reactions were incubated at 37˚C for 10 min, and then, heat-inactivated at 70˚C for 10 min. After DNase I treatment, template DNA was removed by DNA-free DNA Removal kit (Thermo Fisher Scientific, Grand Island, New York, USA). The resulting materials were used as template to generate cDNA using 100 U M-MLV reverse transcriptase (Promega, Madison, Wisconsin, USA), 0.45 μM N6 random hexamer primers (Thermo Fisher Scientific, Grand Island, New York, USA), and 20 U RNase inhibitor (Promega, Madison, Wisconsin, USA). The cDNA was synthesized by incubating reaction at 42˚C for 30 min. Relative mRNA quantitation was done for *cydA* gene using the SYBR green qRT-PCR master mix (Roche, Basel, Switzerland) using specific primers (**Table c in S1 Text**). Data evaluation of 3 biological replicates done in duplicates or triplicate was performed using the threshold cycle ($2^{-\Delta\Delta CT}$) method. qRT-PCR was performed for *gapA* gene using specific primers and probes (**Table c in S1 Text**) containing 5′ 6-carboxyfluorescein and 3′ black-hole quencher 1 modification in a CFX connect Real-Time System (Bio-Rad). PCR-amplified DNA fragments containing the gene of interest were used to generate standard curves.

### Transcriptional pausing

Transcriptional pause assays were performed using PCR-amplified 300 bp promoter with 50 bp coding sequence of either *gapA*, *eno*, *cydA*, or *dksA* gene containing *rrnB* and *rpoC* terminator at the 3′-end. Transcriptional forks were initiated in standard transcription buffer (40 mM HEPES (pH 7.4), 2 mM $MgCl_2$, 60 mM potassium glutamate, 0.1% Nonidet P-40) containing 8 U RiboLock RNase inhibitor, 10 nM RNAP holoenzyme, 2 nM template DNA with or without 200 nM of GreA, 100 nM GreB, or 1 μM DksA proteins. The reactions were carried out for 9 min at 37˚C. Multiround runoff transcription assays were started upon the addition of 200 μM NTPs containing 0.2 μCi [$^{32}$P]-α-UTP (3,000 Ci/mmol). Aliquots removed at between 1 and 9 min were treated with 125 μl of transcription stop buffer (0.6 M Tris-HCl (pH 8.0) and 20 mM EDTA (pH 8.0)) containing 5 μg tRNA. Samples were precipitated with 3 volumes of 100% ethanol, followed by centrifugation at 12,000 rpm for 20 min. The RNA products dissolved in 2× formamide RNA sample buffer were separated in 7 M urea-16% PAGE gels. Transcriptional pause products were identified by using $^{32}$P-labeled Decade Markers System (Ambion) and visualized by the Typhoon PhosphorImager (GE Healthcare).

### Statistical analysis

Statistical analyses were performed using GraphPad Prism 5.0 software. One-way and two-way ANOVA, *t* tests, and logrank tests were used. Data were considered statistically different when $p < 0.05$.

### Supporting information

**S1 Table.** Table A in S1 Table. Transcript error counts. Table B in S1 Table. Counts per transcript.
(XLSX)

**S2 Table.** Table A in S2 Table. RNA seq Δ*greAB* vs. WT. Table B in S2 Table. RNA seq $H_2O_2$ Δ*greAB* vs. $H_2O_2$ WT.
(XLSX)

**S1 Text.** Table a in S1 Text. Bacteria used in this study. Table b in S1 Text. Plasmids used in this study. Table c in S1 Text. Oligonucleotides used in this study. Fig a in S1 Text. KEGG

pathway overrepresentation analysis of SNSs. Fig b in S1 Text. Effect of carbon source on *Salmonella* growth. Fig c in S1 Text. Amino acid pools in *Salmonella* grown on glucose. Fig d in S1 Text. RNA seq analysis of *Salmonella* grown in glucose. Fig e in S1 Text. Resolution of transcriptional pausing by proteins that bind to the secondary channel of RNA polymerase. Fig f in S1 Text. Susceptibility of aerobic respiration mutants to peroxide stress.
(DOCX)

**S1 Data. Data from which the figures were made.**
(XLSX)

**S1 Raw Images. Uncropped images for blots used in the manuscript.**
(PDF)

## Acknowledgments

We thank Dr. Jessica Jones-Carson for kindly providing the mice. We also thank Ted R. Shade from the Genomics Shared Resource Facility, University of Colorado Anschutz Medical Campus, for sequencing of the RNA Seq library, Michael Armstrong Mass Spectrometry Facility, Skaggs School of Pharmacy and Pharmaceutical Sciences, University of Colorado Denver–Anschutz Medical Campus, for analysis of amino acids, and Dr. Tonya Brunetti at the Department of Immunology and Microbiology for her guidance representing and preparing sequencing datasets for publication.

## Author Contributions

**Conceptualization:** Sashi Kant, Andres Vazquez-Torres.

**Formal analysis:** Sashi Kant, James Karl A. Till, Alyssa Margolis, Siva Uppalapati, Andres Vazquez-Torres.

**Funding acquisition:** Andres Vazquez-Torres.

**Investigation:** Sashi Kant, Lin Liu.

**Methodology:** Alyssa Margolis.

**Project administration:** Andres Vazquez-Torres.

**Resources:** Ju-Sim Kim.

**Supervision:** Andres Vazquez-Torres.

**Writing – original draft:** Sashi Kant, Andres Vazquez-Torres.

**Writing – review & editing:** Andres Vazquez-Torres.

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
