## [Editor Report · Decision Letter 0]

26 Aug 2022

Dear Dr Vazquez-Torres, 

Thank you for submitting your manuscript entitled "Rescue of transcriptional pausing in metabolic genes jumpstarts Salmonella antioxidant defenses" for consideration as a Research Article by PLOS Biology. I am handling your manuscript for my colleague Paula whilst she is out of the office. Please accept my apologies for the delay in getting back to you as we consulted with an academic editor about your submission.

Your manuscript has now been evaluated by the PLOS Biology editorial staff, as well as by an academic editor with relevant expertise, and I am writing to let you know that we would like to send your submission out for external peer review.

Once your full submission is complete, your paper will undergo a series of checks in preparation for peer review. After your manuscript has passed the checks it will be sent out for review. To provide the metadata for your submission, please Login to Editorial Manager (https://www.editorialmanager.com/pbiology) within two working days, i.e. by Aug 28 2022 11:59PM.

Kind regards,

Richard

Richard Hodge, PhD

Associate Editor, PLOS Biology

rhodge@plos.org

On behalf of:

Paula Jauregui, PhD

---

## [Decision Letter · Decision Letter 1]

4 Nov 2022

Dear Dr. Vazquez-Torres,

Thank you for your patience while your manuscript "Rescue of transcriptional pausing in metabolic genes jumpstarts Salmonella antioxidant defenses" was peer-reviewed at PLOS Biology. Your manuscript has been evaluated by the PLOS Biology editors, an Academic Editor with relevant expertise, and by several independent reviewers.

As you will see in the reviewer reports, which can be found at the end of this email, although the reviewers find the work potentially interesting, they have also raised a substantial number of important concerns. Based on their specific comments and following discussion with the Academic Editor, it is clear that a substantial amount of work would be required to meet the criteria for publication in PLOS Biology. However, given our and the reviewer interest in your study, we would be open to inviting a comprehensive revision of the study that thoroughly addresses all the reviewers' comments. Given the extent of revision that would be needed, we cannot make a decision about publication until we have seen the revised manuscript and your response to the reviewers' comments. Your revised manuscript would need to be seen by the reviewers again, but please note that we would not engage them unless their main concerns have been addressed. 

Having discussed the reviews with the Academic Editor, we think that it is essential that you improve the writing keeping in mind a broader audience, as all the reviewers have indicated, experimentally address the first two points raised by reviewer 2 that point towards an alternative explanation of the data, and perform the RNAseq experiment suggested by reviewer 3.

We appreciate that these requests represent a great deal of extra work, and we are willing to relax our standard revision time to allow you 6 months to revise your study. Please email us (plosbiology@plos.org) if you have any questions or concerns, or envision needing a (short) extension.

**IMPORTANT - SUBMITTING YOUR REVISION**

*Resubmission Checklist*

*Published Peer Review*

*PLOS Data Policy*

*Blot and Gel Data Policy*

Sincerely,

Paula

---

Senior Editor

PLOS Biology

REVIEWS:

Reviewer #1: Gene regulation.

Reviewer #2: Salmonella gene regulation.

Reviewer #3: Salmonella pathogenesis.

Reviewer #1: In this study, Kant et al. revealed a novel transcriptional mechanism by which Salmonella reprograms its metabolism for protection against oxidative stress. This mechanism is mediated by the GreA and GreB proteins, which resolve transcriptional pausing at genes for EMP (Embden-Meyerhof-Parnas) glycolysis and for ETC (Electron Transport Chain). The authors demonstrate that this regulation protects Salmonella from the cytotoxicity of phagocyte NADPH oxidase in the innate host response. The findings reported by Kant et al. show a role for GreA and GreB transcriptional elongation factors in promoting bacterial growth in a specific stress condition. The manuscript is excellently written, technically sound, conclusions are fully supported by data, and findings are highly relevant in the biology field. 

Comment.

The Figures are blurred.

Reviewer #2: First I must sincerely apologize to the authors, reviewers and editors for being several days late on this review.

The paper by Kant et al. explores the roles of the GreA and GreB factors in the adaptation of Salmonella to oxidative stress. It has a lot of very intriguing data and some important insights. I'm excited by the general trajectory of the hypothesis and its novelty. 

The paper never really comes full circle to answer the following important questions:

- are GreAB really controlling the expression of metabolic genes (directly) under oxidative stress, or is this just a coincidental defect that occurs in a highly pleiotropic set of mutants? The general thrust of the narrative is that GreAB somehow fix pausing during oxidative stress but whether this pausing occurs at higher rates during oxidative stress remains unclear (if I missed some important data I must apologize...I just can't figure out where this might have been addressed). The only control for specificity seems to be the DksA control. It seems to make sense that oxidized nucleotides (or a decrease in nucleotide avaiability) might cause more aberrant and backtracking transcripts to form during elongation. The fact that the pentose phosphate pathway is impacted also seems to suggest that nucleotides might be in short supply (or the cell THINKS they're in short supply)? Although, of course, that pathway also controls the levels of NADPH used for most reductive/anabolic forms of metabolic pathways...so it's a complex question. But I'm simply not clear on why oxidative stress might increase the need for GreAB unless there's a corresponding increase in bad transcripts being formed during elongation. 

- I guess I'm asking if an alternative model could be that oxidative stress drains metabolites from the pentose phosphate pathway away from nucleotide generation...this causes pausing during transcription...which needs GreAB to fix.

- given that over 1000 genes are differentially expressed between the wt and GreAB mutants it's hard to know where the 'specificity' is for these GreAB factors. The levels of many of those transcripts could be regulated indirectly. 

- In looking at where pausing is occurring in the three transcripts analyzed...is there any sequence motif or nucleotide bias that causes RNAP to stall in a Gre-dependent fashion? Are these regions overrepresented in metabolic transcripts and absent in DksA?

- If we grew GreAB cells anaerobically would their transcriptomes be closer to wild-type grown anaerobically as well (would the differences disappear without oxygen)?

Later the paper shifts away from the central narrative of GreAB and tests a wide variety of metabolic pathways and their response to oxidative stress. Some of the findings are less novel but the methylglyoxylate pathway was something I had not read about in this context and this data is quite intriguing. However much of the data is followed up by speculation as to what is happening to the cell. The extra explorations make more noise but don't really help us pin anything down to a clear model. The lack of certain amino acids is not well explained in terms of the larger model, for example. Neither is a link to the stringent response. 

small things:

- the lower amounts of peroxides in GreAB cells isn't paradoxical if they express a lot more catalase (line 109 in text)

- if the nuo/ndh mutants didn't grow on glucose (figure 7) then how was their ability to make peroxide assessed in figure 1?

- figure 7A...should grow in a control media without glucose (or with amino acids?) - are these growth defects specific to gluco-lytic growth or a general overall defect?

So the paper starts out discussing stringent and GreAB and then veers a bit off that track and doesn't really settle in on a specific mechanism but rather tries to explain a broad range of disparate observations that don't always seem connected. I'm sympathetic because these metabolism questions are very hard to get to the bottom of. Very hard! But with respect to GreAB I'm left wondering how much of this is simply a consequence of hitting a pleotropic set of factors and how much of this antioxidant activity is the real role of GreAB factors. 

Reviewer #3: The manuscript entitled: "Rescue of transcriptional pausing in metabolic genes jumpstarts Salmonella antioxidant defenses" by Kant et al describes experiments that attempt to link the Gre factors to metabolism that combats ROS stress in Salmonella. The manuscript is intriguing, but it suffers from a writing style that is confusing and the work has some serious limitations. 

A lot of space in the introduction is given to discussions of DksA, but it is not really important for this paper. Thus, it represents a distracting diversion. The clipped writing style also makes it difficult for appeal to a broad audience. The major rationale appears to be that Gre factors act similarly to DksA, so do they have a role in antioxidant defense? But what the authors actually establish is the role of Gre in metabolism.

This manuscript reads like a series of one-liners strung together. It is extremely disjointed. For example in the experiment shown in Fig. 1F, the authors state that greAB produced significantly more catalase activity without any explanation and suggest that the Gre factors resist oxidative stress in the innate host response by a mechanism that is independent from classical detoxification by catalases. They do not seem to explore this further but drop it out there without explaining what "classical" detoxification by catalases is for comparison. This type of writing is evident throughout the manuscript. 

Major criticisms are enumerated below. 

1. The authors immediately launch into an investigation of virulence in the mouse and the effect of eliminating greA, greB or both gre genes. A very difficult to follow section is then presented where the authors examine nos2-/- or nox2-/- mice. They provide too little rationale for the experiment and fail to state the significance of their findings. This part is written for a very specialized audience and not enough information is provided to the reader. It is clear that greAB is severely attenuated, also in the nox2 knockout, but not in the nos2 knockout. The authors start out with virulence in the mouse, rather than first establishing the effects of gre deletion in Salmonella. It is very sensitive to H2O2 stress (Fig 1C, D). In Fig. 1E, F the legends are woefully inadequate to describe the experiment.

2. Then they move on to the effect of Gre and metabolism. It isn't clear what they mean by "grew poorly", because although in Fig. 1D the gre null strain has a slightly longer lag, it grows to higher density than the wt. Yet the authors do not comment on this. Why? Similarly in the presence of H2O2. In Fig. 2A anaerobic/GLC the authors need to extend the time points further to make their claim. Addition of GSH partially ameliorated the growth defect of the greAB null strain. Again, the writing style is a collection of lines centered around the figure rather than developing the narrative. STATE WHY THE ADDITION OF GSH IS SIGNIFICANT. The appropriate header for this section should be Gre factor mutants suffer from amino acid bradytrophy. ppGpp levels are up about 30%, is this enough to be functionally/physiologically significant? The authors do not relate this finding back to the biology. 

3. In figure 3, the actual data are buried in the supplementary information (S3-4) and should be shown. It isn't clear what percentage of the overall transcriptome involves metabolism, since the authors only validated a subset of genes. But since they go on to further study the ones in the supplementary materials, the data should be in the main text. Why did the authors mention pykF, when it barely meets the threshold of 2 (and only because of an outlier)? The existing Figure 3 can be much smaller and moved to supplemental. 

4. A critical experiment that is missing here is an RNAseq analysis of the Salmonella response to ROS and how it compares to the greAB response. Does the methylgyoxal pathway show up? This would validate the findings of Fig. 6. Again, in this section the authors introduce another pathway and more components (gloB) without a proper introduction of the pathway. The authors conclude that formation of GSH through the cleavage of S-lactoyl-glutathione contributes to antioxidant defenses, but this is independent of greA/B (fig 6F), so why include here? Further, the authors then look at a role for gloB in virulence and include competitive infections, but do not thoroughly describe this figure. This seems like an aside, and it muddies the manuscript. 

5. Then in the last part (Figure 7), the authors go back to redox, and this part is completely disconnected from greA/B. The authors do not connect the dots and it leaves the reader hanging and confused. 

6. I think the authors should consider writing a paper that describes the role of Gre factors in Salmonella metabolism. Later work might be able to link it to ROS stress, but requires a lot more work to establish direct effects. Going back and forth between ROS, Gre, Metabolism and virulence is too confusing and is incomplete in its present form.

Minor comments:

1. line 87 should be an increase in Salmonella fitness or increased Salmonella fitness

2. Figure 1 is too crowded and the y-axes are so compressed that it is difficult to separate the strains in Fig. 1A especially.

3. line 114 should be growth; line 115 should be grew

4. What was the pH of the growth media? I cannot find it mentioned in Figure legends or methods.

5. there are lots of examples of poor English throughout, including supplementary figure legends. Line 145: "because of greAB Salmonella grow poorly" this doesn't make sense.

6. The figures are very difficult to read with small green text that fades into a gray background.

---

## [Decision Letter · Decision Letter 2]

15 Feb 2023

Dear Dr. Vazquez-Torres,

Thank you for your patience while we considered your revised manuscript "Transcription elongation and fidelity Gre factors jumpstart Salmonella oxidative stress resistance" for publication as a Research Article at PLOS Biology. This revised version of your manuscript has been evaluated by the PLOS Biology editors, the Academic Editor, and the original reviewers.

Based on the reviews and on our Academic Editor's assessment of your revision, we are likely to accept this manuscript for publication, provided you satisfactorily address the remaining points raised by the reviewers. We consider that addressing point 5 from reviewer #3, examining why there is a slight difference in the activity of Gre proteins in E. coli and Salmonella, as well as the main concern from reviewer #3 regarding the "how" is not necessary for publication. We consider that this manuscript addresses an important question, i.e. how does a pathogen adapt its metabolism to cope with inflammatory reactive oxygen species (ROS) stress. This manuscript provides a mechanistic link between ROS and reorganization of metabolism through these Gre proteins. We consider that the concern from reviewer #3 is a follow-up question, i.e. to better define the biochemical activities of these Gre proteins, and the current manuscript provides a rationale for these follow-up studies. We think that the findings reported here are important because they reveal how Gre proteins coordinate protection of carbon/energy metabolism during ROS stress, which was unknown. We think that you should address the remaining comments from reviewer #2 and the comments from reviewer #3 regarding the title as there is no evidence for "jump starting", and to tone down superlatives like "pivotal breakpoint". We think that you can consider the rest of the comments from reviewer #3 regarding the presentation of the data, if you think it would improve the clarity. 

Please also make sure to address the following data and other policy-related requests.

1. ETHICS STATEMENT:

-- Please include the full name of the IACUC/ethics committee that reviewed and approved the animal care and use protocol/permit/project license. Please also include an approval number.

-- Please include the specific national or international regulations/guidelines to which your animal care and use protocol adhered. Please note that institutional or accreditation organization guidelines (such as AAALAC) do not meet this requirement.

2. Please provide a blurb which (if accepted) will be included in our weekly and monthly Electronic Table of Contents, sent out to readers of PLOS Biology, and may be used to promote your article in social media. The blurb should be about 30-40 words long and is subject to editorial changes. It should, without exaggeration, entice people to read your manuscript. It should not be redundant with the title and should not contain acronyms or abbreviations.

3. We suggest a change in the title, but please modify as you consider necessary: "Gre factors help Salmonella adapt to reactive oxygen species (ROS) stress by improving transcription elongation and fidelity". 

We expect to receive your revised manuscript within two weeks. 

*Published Peer Review History*

*Press*

Sincerely,

Paula

---

Senior Editor,

pjaureguionieva@plos.org,

PLOS Biology

Reviewer remarks:

Reviewer #1: Víctor H. Bustamante. Gene regulation.

Reviewer #2: Salmonella gene regulation.

Reviewer #3: Salmonella pathogenesis.

Reviewer #1: This new version is event better that the previous manuscript. I congratulate the authors for their excellent study.

Reviewer #2: This revision is a substantial improvement over the prior submission. I think the incorporation of the RNAseq/transcript fidelity data and the streamlined presentation made it a much better read and a more complete story. The authors ditched some data that was very distracting to the central theme and strengthened other aspects. I believe this was the correct choice and that some of their other earlier findings probably belong in a separate publication.

I do think the story never really nails what GreAB is actually doing (which transcripts are targeted and why...signature mutations and motifs were not identifiable although I appreciate the authors looked and stuck to reporting the facts) and I do have some worries that most of these phenomena are side-effects of a highly pleiotropic mutant rather than direct evidence that there is a metabolic 'switch' controlled by GreAB that is activated in response to oxidative stress.

That said there's a lot of novelty here and the experiments are pretty well controlled. I believe the paper will be well read and cited and that this work is a valuable contribution to the field.

I have a few minor comments:

- given that peroxidases and catalases could account for consumption of H2O2, I think the assertion that 'catalase' activity is higher in GreAB mutant cells should be reworded. (line 115) Also I think they may be citing Fig 1F when they mean 1G?

- the extended lag phase in dicarboxylic acids like succinate has been described for salmonella by Hinton's group, Navarre's group (both groups in the past couple of years) and earlier work from Herb Shellhorn. (line 161) Probably should cite one of these.

And some editorializing:

- line 25 = what is an 'emergent aspect'? Keep it simple.

- line 27 - is the stringent response (just) a 'metabolic program'? It also shuts down ribosomal and tRNA synthesis.

- Figure 6I where the authors show that the bd (cydAB) system is required for resistance to oxidative stress...seems very novel and important? It barely gets a mention, whereas I think it should be in the abstract.

Reviewer #3: The manuscript "Transcription elongation and fidelity Gre factors jumpstart Salmonella oxidative stress resistance" by Kant et al is a revised manuscript. The manuscript has been substantially revised and the writing style has improved enormously. However, there are still some fundamental flaws as described below. 

1. The title should be changed, the authors never described how Gre "jumpstarts". The running title is more appropriate.

2. Line 38 what is meant by a "pivotal breakpoint"? It isn't clear.

3. The bottom line here is that Gre factors help reprogram Salmonella metabolism in response to oxidative stress. The manuscript is largely a study of Salmonella metabolism and thus, it lacks broad general interest 

4. L120 the header is completely uninformative. This exemplifies the problems with the writing style that make it difficult for a general audience. The header should be "the absence of gre factors does not exacerbate a decrease in transcriptional fidelity from peroxide stress" or something related. The reader is forced to wade through the entire paragraph to grasp the point. Similarly, and as a general comment, it is impossible to simply look at the figures and follow the thread, because the manuscript contains extraneous data that should be in supplementary and some data in supplementary should be in the main text. 

5. Lines 137-139 the authors state: "The lack of substitution error type bias between untreated wildtype and greAB Salmonella conflicts with prior work in greAB E. coli that identified a strong G>A substitution bias [34, 35]. The authors suggest why this discrepancy might exist but do not examine experimentally. 

6. Lines 120-146 and Figure 2 are essentially negative data, as such, they detract and distract from the manuscript and should be moved to supplementary results. The conclusions: there is no specific SNS type that is inherently driving the overall transcription error rates in wildtype compared to greAB Salmonella, suggesting that in the absence of Gre factors, Salmonella experience nutritional shortages.

7. Since the authors further study gapA, eno and cysA, Figure 3 should emphasize these specific genes and their expression in WT vs gre mutants in both ROS +/- conditions. Thus, in point 4 by reviewer 3 "A critical experiment that is missing here is an RNAseq analysis of the Salmonella response to ROS and how it compares to the greAB response" has not been fully addressed. The Table in supplemental should be moved to the main body.

8. Their observations suggests that Gre factors resolve transcriptional errors in transcripts encoding metabolic functions in Salmonella experiencing oxidative stress. The authors propose that Gre factors regulate assimilation of a variety of glycolytic sugars as well as various carbon sources that enter the TCA, allowing for the balanced production of amino acids. It is a pretty obvious conclusion.

9. Differential expression analyses revealed changes in the transcription of 1,569 genes between greAB and wildtype Salmonella of which 944 were underexpressed and 623 were overexpressed in greAB Salmonella. Gre-dependent rescue of transcriptional pauses is an important step in the activation of key glycolytic genes in Salmonella. The major conclusion is that these investigations indicate that transcriptional control Gre factors exert on ETC genes fosters aerobic metabolism, thereby helping Salmonella meet their energetic and redox needs, while useful information, is hardly novel. The authors never get to the "How?"

---

## [Editor Report · Decision Letter 3]

24 Feb 2023

Dear Dr Vazquez-Torres,

Thank you for the submission of your revised Research Article "Gre factors help Salmonella adapt to oxidative stress by improving transcription elongation and fidelity of metabolic genes" for publication in PLOS Biology. On behalf of my colleagues and the Academic Editor, Sebastian Winter, I am pleased to say that we can in principle accept your manuscript for publication, provided you address any remaining formatting and reporting issues. These will be detailed in an email you should receive within 2-3 business days from our colleagues in the journal operations team; no action is required from you until then. Please note that we will not be able to formally accept your manuscript and schedule it for publication until you have completed any requested changes.

PRESS

Sincerely, 

Paula 

---

Senior Editor

PLOS Biology
